# Global Convergence of Multi-Agent Policy Gradient in Markov Potential Games

**Stefanos Leonardos**
Singapore University of Technology and Design
stefanos_leonardos@sutd.edu.sg

**William Overman**
University of California, Irvine
overmana@uci.edu

**Ioannis Panageas**
University of California, Irvine
ipanagea@ics.uci.edu

**Georgios Piliouras**
Singapore University of Technology and Design
georgios@sutd.edu.sg

## Abstract

Potential games are one of the most important and widely studied classes of normal-form games. They define the archetypal setting of multi-agent coordination in which all agents utilities are perfectly aligned via a common potential function. Can we embed this intuitive framework in the setting of Markov games? What are the similarities and differences between multi-agent coordination with and without state dependence? To answer these questions, we study a natural class of Markov Potential Games (MPGs) that generalizes prior attempts to capture complex stateful multi-agent coordination. Counter-intuitively, insights from normal-form potential games do not carry over since MPGs involve Markov games with zero-sum state-games, but Markov games in which all state-games are potential games are not necessarily MPGs. Nevertheless, MPGs showcase standard desirable properties such as the existence of deterministic Nash policies. In our main result, we prove convergence of independent policy gradient and its stochastic counterpart to Nash policies at a rate that is polynomial in the approximation error by adapting single-agent gradient domination properties to multi-agent settings. This answers questions on the convergence of finite-sample, independent policy gradient methods beyond settings of pure conflicting or pure common interests.

## 1 Introduction

Multi-agent reinforcement learning (MARL) has been the fundamental driver of numerous recent advances in Artificial Intelligence (AI) and Machine Learning (ML) ranging from super-human performance in competitive game-playing (Silver et al., 2016; 2018; Brown & Sandholm, 2019; Jaderberg et al., 2019) and multi-tasking (Mnih et al., 2015; OpenAI, 2018; Vinyals et al., 2019) to robotics, autonomous-driving and cyber-physical systems (Busoniu et al., 2008; Zhang et al., 2019). However, despite the popularity of MARL algorithms to analyze these systems in practice, the theory that underpins their empirical success lags behind (Dafoe et al., 2020). Many state-of-the-art theoretical results concern single-agent RL systems, typically modelled as single-agent Markov Decision Processes (MDPs) (Bertsekas, 2000; Panait & Luke, 2005; Sutton & Barto, 2018).

The main challenge when transitioning from single to multi-agent RL settings is the computation of *Nash policies*. For $n \geq 2$ agents, a Nash policy is defined to be a profile of policies $(\pi_1^*, ..., \pi_n^*)$ so that by fixing the policies of all agents but $i$, $\pi_i^*$ is optimal for the resulting single-agent MDP and this is true for all $1 \leq i \leq n$ [1] (Definition 1). In multi-agent settings, Nash policies *may not be unique* in principle and, unlike singe-agent MDPs, agents' rewards may differ dramatically between them.

A common approach to compute Nash policies in MDPs is the use of *policy gradient* methods. The significant progress in the analysis of such methods (see Agarwal et al. (2020) and references therein) has mainly concerned the single-agent case or the case of pure common interests (identical agents) (Wang & Sandholm, 2002; Panait & Luke, 2005): the convergence properties of policy gradient in

---

[1] Analogue of Nash equilibrium notion.

general MARL settings remain poorly understood. Recent steps towards a theory for multi-agent settings involve Daskalakis et al. (2020) who show convergence of *independent policy gradient* to the optimal policy for two-agent zero-sum stochastic games, Wei et al. (2021) who improve the result of Daskalakis et al. (2020) using optimistic policy gradient and Zhao et al. (2021) who study extensions of Natural Policy Gradient using function approximation. It is worth noting that the positive results of Daskalakis et al. (2020); Wei et al. (2021) and Zhao et al. (2021) depend on the fact that two-agent stochastic zero-sum games satisfy the *min-max equals max-min* property (Shapley, 1953).

If we move away from the extremes of single-agent or purely competitive settings (two-agents, zero-sum), a lot of these regularities, and in particular the *value-uniqueness* property, cease to hold. However, building a theory to analyze *problems of cooperation* between two or more agents constitutes a primary open challenge for the fields of AI and ML (Dafoe et al., 2020; Dafoe et al., 2021). Based on the above, our work is motivated by the following natural question:

> *Can we get (provable) convergence guarantees for multi-agent RL settings*
> *in which agents have aligned incentives, i.e., in which coordination is desirable?*

**Model and Informal Statement of Results.** To make progress in this direction, we study a class of $n$-agent MDPs that naturally generalize normal form potential games (Monderer & Shapley, 1996), the archetypal model of interactions between multiple agents with aligned, yet not necessarily identical interests, called *Markov Potential Games (MPGs)*. In words, a multi-agent MDP is an MPG as long as there exists a (state-dependent) real-valued potential function $\Phi$ so that if an agent $i$ changes their policy (and the rest of the agents keep their policy unchanged), the difference in agent $i$'s value/utility, $V^i$, is captured by the difference in the value of $\Phi$ (Definition 2).

Our first task is to understand the structural properties of MPGs and their Nash policies. Rather surprisingly, many insights from normal-form potential games do not carry over as MPGs involve settings with purely competitive (zero-sum) interactions at some states. Moreover, Markov games in which every state-interaction is a potential game are not necessarily MPGs. These findings suggest that MPGs form a class of MDPs with rich structure which challenges our intuition on the nature of cooperation in state-based interactions. On the other hand, MPGs trivially include MDPs of pure common interests (MDPs in which agents have identical rewards) and showcase intuitively expected properties such as the existence of deterministic Nash policies. Our structural results are as follows.

**Theorem 1.1** (Structural Properties of MPGs)**.** *The following facts are true for MPGs with $n$-agents.*
*(a) There always exists a deterministic Nash policy profile (see Theorem 3.1).*
*(b) We can construct MDPs for which each state is a (normal-form) potential game but which are not MPGs. This can be true regardless of whether the whole MDP is competitive or cooperative in nature (see Examples 1 and 2, respectively). On the opposite side, we can construct MDPs that are MPGs, but which include states that are purely competitive (i.e., zero-sum games), see Figure 3.*
*(c) We provide sufficient conditions so that an MDP is an MPG. These include cases where each state is a (normal-form) potential game and the transition probabilities are not affected by agents actions or the reward functions satisfy certain regularity conditions between different states (see conditions C1 and C2 in Proposition 3.2).*

We then turn to our motivating question above and, in our main contribution, we answer it in the affirmative. We show that if every agent $i$ independently runs (with simultaneous updates) projected gradient ascent (PGA) on their policy (using their value $V^i$), then, after $O(1/\epsilon^2)$ iterations, the system will reach an $\epsilon$-approximate Nash policy. Here, independence means that (PGA) requires only local information to determine the updates, i.e., each agent's own rewards, actions, and view of the environment. Such protocols are naturally motivated in distributed settings where all information about type of interaction and other agents' actions is encoded in the agent's environment. For the finite samples analogue, we show that the system will reach an $\epsilon$-approximate Nash policy after $O(1/\epsilon^6)$ iterations. Our main convergence results are summarized in the following (informal) Theorem.

**Theorem 1.2** (Convergence of Policy Gradient (Informal))**.** *Consider an MPG with $n$ agents and let $\epsilon > 0$.  (a)* Exact Gradients: *If each agent $i$ runs independent policy gradient using direct parameterization on their policy and the updates are simultaneous, then the learning dynamics reach an $\epsilon$-Nash policy after $\mathcal{O}(1/\epsilon^2)$ iterations. (b)* Finite Samples: *If each agent $i$ runs stochastic policy gradient using greedy parameterization (see equation 3) on their policy and the updates are simultaneous, then the learning dynamics reach an $\epsilon$-Nash policy after $\mathcal{O}(1/\epsilon^6)$ iterations.*

The formal statements for cases (a) and (b) are provided in Theorems 4.2 and 4.4, respectively. The technical details are presented in Section 4. The main step in the proof of Theorem 1.2 establishes that Projected Gradient Ascent (PGA) on the potential function generates the same dynamics as each agent running independent PGA on their value function. This follows from a straightforward derivation of an agent-wise version of the single-agent gradient domination property which can be used to show that every *(approximate) stationary point* (Definition 4) of the potential function is an *(approximate) Nash policy* (Lemma 4.1). If agents do not have access to exact gradients, the key is to get an *unbiased sample* for the gradient of the value functions and prove that it has bounded variance (in terms of the parameters of the MPG). This is established by requiring agents to perform stochastic PGA with $\alpha$-greedy exploration (equation 3). The main idea is that this parameterization stays away from the boundary of the simplex throughout its trajectory (Daskalakis et al., 2020).

**Other Related Works.** Our paper contributes to the growing literature on cooperative AI and ML (Carroll et al., 2019; Dafoe et al., 2020). The results on convergence of MARL algorithms are scarce and largely restricted to purely competitive (Daskalakis et al., 2020; Wei et al., 2021; Zhao et al., 2021) or purely cooperative (Wang & Sandholm, 2002; Bard et al., 2020) settings. As Daskalakis et al. (2020) argue, the current frontier concerns the extension to settings that are not zero-sum, involve more than two agents and/or are cooperative in nature, albeit likely for weaker solution concepts. Our current paper proceeds precisely in this direction, and in fact, it does so without reverting to a weaker solution concept. Concerning the setup, our paper contributes to the rapidly growing literature on MPGs and variations thereof which is showcased by the works of Marden (2012); Valcarcel Macua et al. (2018); Mguni (2021); Mguni et al. (2021) and the partially concurrent work of Zhang et al. (2021). Marden (2012) study Markov games that are potential at every state and which satisfy a strong additional *state-transitivity* property. Under the same assumption, Mguni (2021) derive an analogous result to our Theorem 3.1 on the existence of deterministic Nash policies. Valcarcel Macua et al. (2018) provide an analytical way to find a closed-loop Nash policy in a class of Markov games that is more closely related to our current MPG setting (cf. Proposition 3.2). Zhang et al. (2021) study the same class of MPGs, and present additional practical applications. By introducing the notion of averaged MDPs, they derive an alternative, model-based policy evaluation method, which, interestingly, establishes the same sample-complexity (of $\mathcal{O}(\epsilon^{-6})$) as our model-free estimation. As pointed out by Zhang et al. (2021), it will be instructive to further explore this connection. Concerning the technical parts, our methods are related to Daskalakis et al. (2020); Agarwal et al. (2020); Kakade & Langford (2002) and to Davis & Drusvyatskiy (2018); Bubeck (2015); Nemirovski et al. (2009).

## 2 PRELIMINARIES

**Markov Decision Processes (MDPs).** We consider a setting with $n$ agents who select actions in a shared Markov Decision Process (MDP). Formally, an MDP is a tuple $\mathcal{G} = (\mathcal{S}, \mathcal{N}, \{\mathcal{A}_i, R_i\}_{i \in \mathcal{N}}, P, \gamma, \rho)$, where $\mathcal{S}$ is a finite state space of size $S = |\mathcal{S}|, \mathcal{N} = \{1, 2, \ldots, n\}$ is the set of agents, and $\mathcal{A}_i$ is a finite action space of size $A_i = |\mathcal{A}_i|$ for each agent $i \in \mathcal{N}$ with generic element $a_i \in \mathcal{A}_i$. We write $\mathcal{A} = \prod_{i \in \mathcal{N}} \mathcal{A}_i$ and $\mathcal{A}_{-i} = \prod_{j \neq i} \mathcal{A}_j$ to denote the joint action spaces of all agents and of all agents other than $i$ with generic elements $\mathbf{a} = (a_i)_{i \in \mathcal{N}}$ and $\mathbf{a}_{-\mathbf{i}} = (a_j)_{j \neq i}$, respectively. $R_i : \mathcal{S} \times \mathcal{A} \to [-1, 1]$ is the reward function of agent $i \in \mathcal{N}$, i.e., $R_i(s, a_i, \mathbf{a}_{-i})$ is the instantaneous reward of agent $i$ when they take action $a_i$ and all other agents take actions $\mathbf{a}_{-i}$ at state $s \in \mathcal{S}$. $P$ is the transition probability function, for which $P(s' \mid s, \mathbf{a})$ is the probability of transitioning from $s$ to $s'$ when $\mathbf{a} \in \mathcal{A}$ is chosen by the agents. Finally, $\gamma$ (same for all agents) is a discount factor for future rewards and $\rho \in \Delta(\mathcal{S})$ is a distribution for the initial state at time $t = 0$.[2]

Whenever time is relevant, we will index the above terms with $t$. In particular, at each time step $t \geq 0$, all agents $i \in \mathcal{N}$ observe the state $s_t \in \mathcal{S}$, select actions $\mathbf{a}_t = (a_{i,t}, \mathbf{a}_{-i,t})$, receive rewards $r_{i,t} := R_i(s_t, \mathbf{a}_t)$, and transition to the next state $s_{t+1} \sim P(\cdot \mid s_t, \mathbf{a}_t)$. We will write $\tau = (s_t, \mathbf{a}_t, \mathbf{r}_t)_{t \geq 0}$ to denote the trajectories of the system, where $\mathbf{r}_t := (r_{i,t}), i \in \mathcal{N}$.

**Policies and Value Functions.** For each agent $i \in \mathcal{N}$, a deterministic, stationary policy $\pi_i : \mathcal{S} \to \mathcal{A}_i$ specifies the action of agent $i$ at each state $s \in \mathcal{S}$, i.e., $\pi_i(s) = a_i \in \mathcal{A}_i$ for each $s \in \mathcal{S}$. A stochastic, stationary policy $\pi_i : \mathcal{S} \to \Delta(\mathcal{A}_i)$ specifies a probability distribution over the actions

---

[2]We will write $\Delta(\mathcal{X})$ to denote the set of probability distributions over any set $\mathcal{X}$.

of agent $i$ for each state $s \in \mathcal{S}$. Accordingly, $\pi_i \in \Pi_i := \Delta(\mathcal{A}_i)^S$. In this case, we will write $a_i \sim \pi_i(\cdot \mid s)$ to denote the randomized action of agent $i$ at state $s \in \mathcal{S}$. As above, we will write $\pi = (\pi_i)_{i \in \mathcal{N}} \in \Pi := \times_{i \in \mathcal{N}} \Delta(\mathcal{A}_i)^S$ and $\pi_{-i} = (\pi_j)_{i \neq j \in \mathcal{N}} \in \Pi_{-i} := \times_{i \neq j \in \mathcal{N}} \Delta(\mathcal{A}_j)^S$ to denote the joint policies of all agents and of all agents other than $i$, respectively. A joint policy $\pi$ induces a distribution $\Pr^\pi$ over trajectories $\tau = (s_t, \mathbf{a}_t, \mathbf{r}_t)_{t \geq 0}$, where $s_0$ is drawn from the initial state distribution $\rho$ and $a_{i,t}$ is drawn from $\pi_i(\cdot \mid s_t)$ for all $i \in \mathcal{N}$.

The value function, $V_s^i : \Pi \to \mathbb{R}$, gives the expected reward of agent $i \in \mathcal{N}$ when $s_0 = s$ and the agents draw their actions, $\mathbf{a}_t = (a_{i,t}, \mathbf{a}_{-i,t})$, at time $t \geq 0$ from policy $\pi = (\pi_i, \pi_{-i})$

$$V_s^i(\pi) := \mathbb{E}_\pi \left[ \sum\nolimits_{t=0}^\infty \gamma^t r_{i,t} \mid s_0 = s \right]. \tag{1}$$

We also write $V_\rho^i(\pi) = \mathbb{E}_{s \sim \rho} \left[ V_s^i(\pi) \right]$ if the initial state is random and follows distribution $\rho$. The solution concept that we will be focusing on are the *Nash Policies* which are formally defined next.

**Definition 1** ($\epsilon$-Nash Policy). A joint policy $\pi^* = (\pi_i^*)_{i \in \mathcal{N}}$ is an $\epsilon$-Nash policy if there exists an $\epsilon \geq 0$ so that for each agent $i \in \mathcal{N}$, $V_s^i(\pi_i^*, \pi_{-i}^*) \geq V_s^i(\pi_i, \pi_{-i}^*) - \epsilon$, for all $\pi_i \in \Delta(\mathcal{A}_i)^S$, and all $s \in \mathcal{S}$. If $\epsilon = 0$, then $\pi^*$ is a called a *Nash policy*. In this case, $\pi_i^*$ maximizes each agent $i$'s value function for each starting state $s \in \mathcal{S}$ given the policies, $\pi_{-i}^* = (\pi_j^*)_{j \neq i}$, of all other agents $j \neq i \in \mathcal{N}$. The definition of a Nash policy remains the same if $s \sim \rho$ (random starting state).

## 3 MARKOV POTENTIAL GAMES.

We are now ready to define the class of MDPs that we will focus on for the rest of the paper.

**Definition 2** (Markov Potential Game). A Markov Decision Process (MDP), $\mathcal{G}$, is called a *Markov Potential Game (MPG)* if there exists a (state-dependent) function $\Phi_s : \Pi \to \mathbb{R}$, with $s \in \mathcal{S}$, so that

$$V_s^i(\pi_i, \pi_{-i}) - V_s^i(\pi_i', \pi_{-i}) = \Phi_s(\pi_i, \pi_{-i}) - \Phi_s(\pi_i', \pi_{-i}),$$

for all agents $i \in \mathcal{N}$, states $s \in \mathcal{S}$ and policies $\pi_i, \pi_i' \in \Pi_i, \pi_{-i} \in \Pi_{-i}$. Linearity of expectation implies that $\Phi_\rho(\pi_i, \pi_{-i}) - \Phi_\rho(\pi_i', \pi_{-i}) = V_\rho^i(\pi_i, \pi_{-i}) - V_\rho^i(\pi_i', \pi_{-i})$, where $\Phi_\rho(\pi) := \mathbb{E}_{s \sim \rho} \left[ \Phi_s(\pi) \right]$.

As in normal-form games, an immediate consequence of this definition is that the value function of each agent in an MPG can be written as the sum of the potential *(common term)* and a term that does not depend on that agent's policy *(dummy term)*, i.e., for each agent $i \in \mathcal{N}$ there exists a function $U_s^i : \Pi_{-i} \to \mathbb{R}$ so that $V_s^i(\pi) = \Phi_s(\pi) + U_s^i(\pi_{-i})$, for all $\pi \in \Pi$ (cf. Proposition B.1 in Appendix B).

**Definition 3** (Ordinal and Weighted Potential Games). Similar to normal-form games, we can define more general notions of MPGs: *weighted* or *ordinal* MPGs. If there exist positive constants $w_i > 0, i \in \mathcal{N}$ so that $w_i \left[ V_s^i(\pi_i, \pi_{-i}) - V_s^i(\pi_i', \pi_{-i}) \right] = \Phi_s(\pi_i, \pi_{-i}) - \Phi_s(\pi_i', \pi_{-i})$, then $\mathcal{G}$ is a *Weighted Markov Potential Game (WMPG)*. If for all agents $i \in \mathcal{N}$, all states $s \in \mathcal{S}$ and all policies $\pi_i, \pi_i' \in \Pi_i, \pi_{-i} \in \Pi_{-i}$, the function $\Phi_s, s \in \mathcal{S}$ satisfies $V_s^i(\pi_i, \pi_{-i}) - V_s^i(\pi_i', \pi_{-i}) > 0 \iff \Phi_s(\pi_i, \pi_{-i}) - \Phi_s(\pi_i', \pi_{-i}) > 0$, then $\mathcal{G}$ is an *Ordinal Markov Potential Game (OMPG)*.

Such classes are naturally motivated in the setting of multi-agent MDPs. As Example 2 shows, even simple potential-like settings, i.e., settings in which coordination is desirable for all agents, may fail to be exact MPGs (but may still be ordinal or weighted MPGs). From our current perspective, ordinal and weighted MPGs remain relevant since our main convergence results on the convergence of policy gradient carry over (in an exact or asymptotic sense) also in these classes of games (Remark 1).

**Existence of Deterministic Nash Policies in MPGs.** Before studying the types of MDPs that are captured by Definition 2, we first show that MPGs always possess deterministic Nash policies (similar to their single-state counterparts, i.e., to normal-form potential games Monderer & Shapley (1996)). This is established in Theorem 3.1, which settles part (a) of Theorem 1.1. As with the rest of the proofs (and technical details) of Section 3, the proof of Theorem 3.1 is provided in Appendix B.

**Theorem 3.1** (Deterministic Nash Policy Profile). *Let $\mathcal{G}$ be a Markov Potential Game (MPG). Then, there exists a Nash policy $\pi^* \in \Delta(\mathcal{A})^S$ which is deterministic, i.e., for each agent $i \in \mathcal{N}$ and each state $s \in \mathcal{S}$, there exists an action $a_i \in \mathcal{A}_i$ so that $\pi_i^*(a_i \mid s) = 1$.*

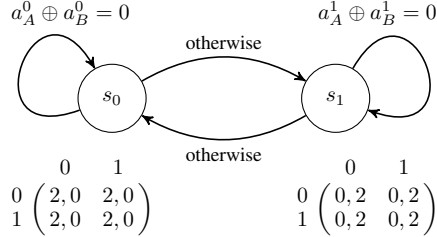 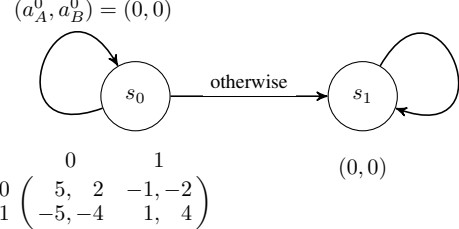

Figure 1: an MDP with normal-from potential games at each state which is not an MPG due to conflicting preferences over states.

Figure 2: an MDP with normal-form potential games at each state which is an OMPG but not an MPG despite common preferences over states.

The proof of Theorem 3.1 relies on the following iterative reduction process of the non-deterministic components of an arbitrary Nash policy profile that is also a global maximizer of the potential function. At each iteration, we isolate an agent $i \in \mathcal{N}$, and find a deterministic (optimal) policy for that agent in the (single-agent) MDP in which the policies of all other agents but $i$ remain fixed. The important observation is that the resulting profile is again a global maximizer of the potential and hence, a Nash policy profile. This argument critically relies on the MPG structure (cf. Definition 2) and does not seem to be directly generalizable to MDPs beyond this class.

**Sufficient Conditions for MPGs.** Based on the above, it is tempting to think that MDPs which are potential at every state (i.e., the immediate rewards at every state are captured by a potential game at that state) are trivially MPGs. As we show in Examples 1 and 2, this intuition fails in the most straightforward way: we can construct simple MDPs that are potential at every state but which are purely competitive overall (do not possess a deterministic Nash policy, see Example 1) or which are cooperative in nature overall, but which do not possess an exact potential function, see Example 2.

**Example 1.** Consider the MDP in Figure 1. To show that $\mathcal{G}$ is not an MPG, it suffices to show that it cannot have a deterministic Nash policy profile as should be the case according to Theorem 3.1 (see Appendix B.1). Intuitively, competition arises in Example 1 because the two agents play a game of *matching pennies* in terms of the states that they prefer (which can be determined by the actions that they choose) even though immediate rewards at each state are determined by normal form potential games. Example 2 shows that a state-based potential game may fail to be an MPG even if agents have similar preferences over states.

**Example 2.** Consider the MDP in Figure 2. In $s_0$ the agents play a Battle of the Sexes game and hence a potential game, while in $s_1$ they receive no reward (which is trivially a potential game). A simple calculation shows that there is not an exact potential function due to the dependence of the transitions on agents' actions (thus, this MDP is not an MPG). However, in the case of Example 2, it is straightforward to show that the game is an ordinal potential game, cf. Appendix B.1.

The previous discussion focuses on games that consist of normal-form potential games at every state, which leaves an important question unanswered: are there games which are not potential at every state but which are captured by the current definition of MPGs? Figure 3 answers this question affirmatively. Together with Example 1, this settles the claim in Theorem 1.1, part (b).

**Proposition 3.2** (Sufficient Conditions for MPGs). *Consider an MDP, $\mathcal{G}$, in which every state $s \in \mathcal{S}$ is a potential game, i.e., the immediate rewards $R(s, \mathbf{a}) = (R_i(s, \mathbf{a}))_{i \in \mathcal{N}}$ for each state $s \in \mathcal{S}$ are captured by the utilities of a potential game with potential function $\phi_s$. If either C1 or C2 are true, then $\mathcal{G}$ is an MPG.*

*C1. Agent-independent transitions: $P(s' \mid s, \mathbf{a})$ does not depend on $\mathbf{a}$, that is, $P(s' \mid s, \mathbf{a}) = P(s' \mid s)$ is just a function of the present state for all states $s, s' \in \mathcal{S}$.*
*C2. Constant gradient of dummy terms: $P(s' \mid s, \mathbf{a})$ is arbitrary, but for the decomposition of each agent's immediate rewards in common and dummy terms, $R_i(s, a_i, \mathbf{a}_{-i}) = \phi_s(a_i, \mathbf{a}_{-i}) + u_s^i(\mathbf{a}_{-i})$ with $u_s^i : \mathcal{A}_{-i} \to \mathbb{R}$, it holds that*

$$\nabla_{\pi_i(s)} \mathbb{E}_{\tau \sim \pi} \left[ \sum_{t=0}^{\infty} \gamma^t u_{s_t}^i(\mathbf{a}_{-i,t}) \mid s_0 = s' \right] = c_s \mathbf{1},$$

*for all states $s', s \in \mathcal{S}$, where $c_s \in \mathbb{R}$, $\mathbf{1} \in \mathbb{R}^{A_i}$, and $\pi_i(s)$ is the policy distribution of agent $i$ at $s$.*

Condition C1 can be viewed as a special case of condition C2; nevertheless, it is more practical to state it separately. Condition C2 (or variations) have been studied in Marden (2012); Valcarcel Macua

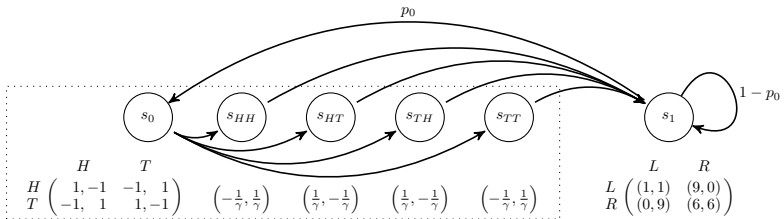

Figure 3: A 2-player MDP which is not potential at every state but which is overall an MPG. While state $s_0$ corresponds to a zero-sum game, the states inside the dotted rectangle do form a potential game which can be used to show the MPG property whenever $p_0$ does not depend on agents' actions.

et al. (2018). However, Example 2 demonstrates that such conditions are restrictive, in the sense that they do not capture very simple MDPs in which agents have aligned incentives (cooperative in nature). Condition C2 is (trivially) satisfied when $u_s^i$ does not depend on the state $s$ nor on the actions of other agents, i.e., $u_s^i(\mathbf{a}_{-i}) \equiv c^i$ for some constant $c^i \in \mathbb{R}$ for all $\mathbf{a}_{-i} \in \mathcal{A}_{-i}$ and all $s \in \mathcal{S}$. A special case is provided by MDPs in which the instantaneous rewards of all agents are the same at each state, i.e., such that $R_i(s, a_i, \mathbf{a}_{-i}) = \phi_s(a_i, \mathbf{a}_{-i})$ for all agents $i \in \mathcal{N}$, all actions $a_i \in \mathcal{A}_i$ and all states $s \in \mathcal{S}$. MDPs that satisfy this condition form a subclass of the current definition and have been studied under the name *Team Markov Games* in Wang & Sandholm (2002). These observations motivate the study of ordinal or weighted MPGs (cf. Definition 3) as more glasses classes of cooperative MDPs.

## 4 CONVERGENCE OF POLICY GRADIENT IN MARKOV POTENTIAL GAMES

In the current section, we present the main results regarding convergence of (projected) policy gradient to approximate Nash policies in Markov Potential Games (MPGs). We analyze the cases of both infinite and finite samples using direct and $\alpha$-greedy parameterizations, respectively. All proofs and auxiliary materials are deferred to Appendix D.

**Independent Policy Gradient and Direct Parameterization.**  We assume that all agents update their policies *independently* according to the *projected gradient ascent (PGA)* or *policy gradient* algorithm.[3] The PGA algorithm is given by

$$\pi_i^{(t+1)} := P_{\Delta(\mathcal{A}_i)^S}\left(\pi_i^{(t)} + \eta\nabla_{\pi_i}V_\rho^i(\pi^{(t)})\right), \tag{PGA}$$

for each agent $i \in \mathcal{N}$, where $P_\Delta(\mathcal{A}_i)^S$ is the projection onto $\Delta(\mathcal{A}_i)^S$ in the Euclidean norm. Here, the additional argument $t \geq 0$ denotes time. We also assume that all players $i \in \mathcal{N}$ use direct policy parameterizations, i.e., $\pi_i(a \mid s) = x_{i,s,a}$, with $x_{i,s,a} \geq 0$ for all $s \in \mathcal{S}, a \in \mathcal{A}_i$ and $\sum_{a \in \mathcal{A}_i} x_{i,s,a} = 1$ for all $s \in \mathcal{S}$. This parameterization is complete in the sense that any stochastic policy can be represented in this class Agarwal et al. (2020).

In practice, agents use *projected stochastic gradient ascent* (PSGA), according to which, the actual gradient, $\nabla_{\pi_i}V_\rho^i(\pi^{(t)})$, is replaced by an estimate thereof that is calculated from a randomly selected (yet finite) sample of trajectories of the MDP. This estimate, $\hat{\nabla}_{\pi_i}^{(t)}$ may be derived from a single or a batch of observations which in expectation behave as the actual gradient. We choose the estimator of the gradient of $V_\rho^i$ to be

$$\hat{\nabla}_{\pi_i}^{(t)} = R_i^{(T,t)}\sum_{k=0}^{T}\nabla\log\pi_i(a_k^{(t)} \mid s_k^{(t)}), \tag{2}$$

where $s_0^t \sim \rho$, and $R_i^{(T,t)} = \sum_{k=0}^{T}r_{i,t}^k$ is the sum of rewards of agent $i$ for a batch of time horizon $T$ along the trajectory generated by the stochastic gradient ascent algorithm at its $t$-th iterate.

The direct parameterization is not sufficient to ensure that the variance of the gradient estimator is bounded (as policies approach the boundary). In this case, we will require that each agent $i \in \mathcal{N}$ uses instead direct parameterization with $\alpha$-greedy exploration as follows

$$\pi_i(a \mid s) = (1-\alpha)x_{i,s,a} + \alpha/A_i, \tag{3}$$

---

[3]In practice, even though each agent treats their environment as fixed, the environment changes as other agents update their policies. This makes the analysis of such protocols particularly challenging in general and highlights the importance of studying classes of MDPs in which convergence can be obtained.

where $\alpha$ is the exploration parameter for all agents. For $\alpha$-greedy exploration, it can be shown that (2) is unbiased and has bounded variance. In this case, $1 - \gamma$ captures the probability that the MDP terminates after each round which ensures that the (finite) length of the trajectory is sampled from a geometric distribution. This is necessary for unbiasedness, see Zhang et al. (2020); Paternain et al. (2021) and Lemma 4.3. (PSGA) is given by

$$\pi_i^{(t+1)} := P_{\Delta(\mathcal{A}_i)^S}\left(\pi_i^{(t)} + \eta\hat{\nabla}_{\pi_i}^{(t)}\right). \tag{PSGA}$$

**Main Technical Tools.** The first step is to observe that, in MPGs, the (partial) derivatives of the value functions and the potential function are equal, i.e., $\nabla_{\pi_i}V_s^i(\pi) = \nabla_{\pi_i}\Phi(\pi)$ for all $i \in \mathcal{N}$ (Proposition B.1). Together with the separability of the projection operator, i.e., the fact that projecting independently for each agent $i$ on $\Delta(\mathcal{A}_i)^S$ is the same as jointly projecting on $\Delta(\mathcal{A})^S$ (Lemma D.1), this establishes that running PGA or PSGA on each agent's value function is equivalent to running PGA or PSGA on the potential function $\Phi$. The next step is to study the stationary points of $\Phi$.

**Definition 4** ($\epsilon$-Stationary Point of $\Phi$). A policy profile $\pi := (\pi_1, ..., \pi_n) \in \Delta(\mathcal{A})^S$ is called $\epsilon$-stationary for $\Phi$ w.r.t distribution $\mu$ as long as $\max \sum_{i \in \mathcal{N}} \delta_i^\top \nabla_{\pi_i}\Phi_\mu(\pi) \leq \epsilon$, where the $\max$ is taken over all $\delta = (\delta_1, \ldots, \delta_n)$ such that $(\pi_1 + \delta_1, \ldots, \pi_n + \delta_n) \in \Delta(\mathcal{A})^S, \sum_{i \in \mathcal{N}} \|\delta_i\|_2^2 \leq 1$. In words, the function $\Phi(\pi)$ cannot increase in value by more than $\epsilon$ along every possible local direction $\delta$ that is feasible (namely $\pi + \delta$ is also a policy profile).

To state our main result, we will also need the notion of *distribution mismatch coefficient* (Kakade & Langford, 2002) applied to our setting. For any distribution $\mu \in \Delta(\mathcal{S})$ and any policy $\pi \in \Pi$, we call $D := \max_{\pi \in \Pi}\left\|\frac{d_\mu^\pi}{\mu}\right\|_\infty$ the *distribution mismatch coefficient*, where $d_\mu^\pi$ is the discounted state distribution (7). Lemma 4.1 suggests that as long as policy gradient reaches a point $\pi^{(t)}$ with small gradient along the directions in $\Delta(\mathcal{A})^S$, it must be the case that $\pi^{(t)}$ is an approximate Nash policy.

**Lemma 4.1** (Stationarity of $\Phi$ implies Nash). *Let $\epsilon \geq 0$, and let $\pi$ be an $\epsilon$-stationary point of $\Phi$ (see Definition 4). Then, it holds that $\pi$ is a $\sqrt{S}D\epsilon(1-\gamma)^{-1}$-Nash policy.*

Lemma 4.1 will be one of two mains ingredients to establish convergence of PGA and PSGA. To prove Lemma 4.1, we will use an agent-wise version of the gradient domination property of single-agent MDPs (cf. Lemma D.3 and Lemma 4.1 in Agarwal et al. (2020)). An important difference in the gradient domination property between (multi-agent) MPGs and single agent MDPs is that the value of each agent at different Nash policies may not be unique[4] which implies that the gradient domination property will only be enough to guarantee convergence to *one of* the optimal (stationary) points of $\Phi$ (and not necessarily to the absolute maximum of $\Phi$).

The second main ingredient will be fact that $\Phi$ is a $\beta$-smooth function (its gradient is Lipschitz) with parameter $\beta = 2n\gamma A_{\max}(1-\gamma)^{-3}$, where $A_{\max} := \max_{i \in \mathcal{N}}|\mathcal{A}_i|$, i.e., the maximum number of actions for some agent. Importantly, this implies that $\beta$ scales *linearly* in the number of agents.

**Exact Gradients Case.** Theorem 1.2 (restated formally below) about rates of convergence of PGA can now be proved following a standard ascent property of gradient descent algorithms to approximate stationary points in non-convex optimization Ghadimi & Lan (2013). The *Ascent Lemma* (Lemma C.1) suggests that for any $\beta$-smooth function, $f$, it holds that $f(x') - f(x) \geq \frac{1}{2\beta}\|x' - x\|_2^2$, where $x'$ is the next iterate of equation PGA. Thus, having shown in our context that $\Phi$ is a $\beta$-smooth function, this implies that

$$\Phi_\mu(\pi^{(t+1)}) - \Phi_\mu(\pi^{(t)}) \geq \frac{(1-\gamma)^3}{4n\gamma A_{\max}}\left\|\pi^{(t+1)} - \pi^{(t)}\right\|_2^2. \tag{4}$$

Putting everything together, we can show the following result.

**Theorem 4.2** (Formal Theorem 1.2, part (a)). *Let $\mathcal{G}$ be an MPG and let $A_{\max} = \max_i |\mathcal{A}_i|$. For any initial state and any initial policies, if all agents run independent projected gradient ascent (PGA) on their policies with learning rate (step-size) $\eta = \frac{(1-\gamma)^3}{2n\gamma A_{\max}}$ for at least $T = \frac{16n\gamma D^2 S A_{\max}\Phi_{\max}}{(1-\gamma)^5\epsilon^2}$ iterations, then there exists a $t \in \{1, \ldots, T\}$ such that $\pi^{(t)}$ is an $\epsilon$-approximate Nash policy.*

---

[4]This is in contrast to single-agent MDPs for which the agent has a unique optimal value even though their optimal policy may not necessarily be unique.

**Finite Sample Case.** In the case of finite samples, we analyze (PSGA) on the value $V^i$ of each agent $i$ which (as was the case for PGA) can be shown to be the same as applying projected gradient ascent on $\Phi$. The key is to get an estimate of the gradient of $\Phi$ (2) at every iterate. Note that $1 - \gamma$ now captures the probability for the MDP to terminate after each round (and it does not play the role of a discounted factor since we consider finite length trajectories). Lemma 4.3 argues that the estimator of equation equation 2 is both unbiased and bounded.

**Lemma 4.3** (Unbiased estimator with bounded variance). *It holds that $\hat{\nabla}_{\pi_i}^{(t)}$ is an unbiased estimator of $\nabla_{\pi_i}\Phi$ for all $i \in \mathcal{N}$, that is $\mathbb{E}_{\pi^{(t)}}\hat{\nabla}_{\pi_i}^{(t)} = \nabla_{\pi_i}\Phi_\mu(\pi^{(t)})$, for all $i \in \mathcal{N}$. Moreover, for all agents $i \in \mathcal{N}$, it holds that $\mathbb{E}_{\pi^{(t)}}\left\|\hat{\nabla}_{\pi_i}^{(t)}\right\|_2^2 \leq \frac{24A_{\max}^2}{\epsilon(1-\gamma)^4}$, for all $i \in \mathcal{N}$.*

Using the above, we can now state part (b) of Theorem 1.2. Together with Lemma 4.3 and the stationarity Lemma (Lemma 4.1), the proof of this part uses the smoothness of $\Phi$ and existing tools from the analysis of stochastic gradient descent for non-convex functions.

**Theorem 4.4** (Formal Theorem 1.2, part (b)). *Let $\mathcal{G}$ be an MPG. For any initial state and any initial policies, if all agents run projected stochastic gradient ascent (PSGA) on their policies using $\alpha$-greedy parameterization with $\alpha = \epsilon^2$ and learning rate (step-size) $\eta = \frac{\epsilon^4(1-\gamma)^3\gamma}{48nD^2A_{\max}^2 S}$ for at least $T = \frac{48(1-\gamma)A_{\max}\Phi_{\max}D^4 S^2}{\epsilon^6\gamma^3}$ iterations, then there exists a $t \in \{1, \ldots, T\}$ such that in expectation, $\pi^{(t)}$ is an $\epsilon$-approximate Nash policy.*

Using Markov's inequality, it is immediate to show that the statement of Theorem 4.4 also holds with high probability. Namely, if we set the number of iterations to $T_\delta = T/\delta^4$ and and the learning rate to $\eta_\delta = \delta^2\eta$, where $\delta \in (0, 1)$, then with probability $1 - \delta$ there exists a $t \in \{1, \ldots, T\}$ such that $\pi^{(t)}$ is an $\epsilon$-approximate Nash policy. However, this is a weaker than desired statement, since, optimally, the running time should be logarithmic (instead of polynomial) in $1/\delta$ (Nemirovski et al., 2009). Proving such a statement requires bounds on higher moments of the gradient estimator which in turn, require sampling and averaging over multiple trajectories (per iteration).

*Remark* 1 (Convergence in WMPGs and OMPGs). It is rather straightforward to see that our results carry over to WMPGs (cf. Definition 3). The only difference in the running time of PGA is to account for the weights (which are just multiplicative constants). By contrast, the extension to OMPGs is not immediate. The reason is that, in that case, we cannot prove any bound on the smoothness of $\Phi$ and hence, we cannot have rates of convergence of policy gradient. Nevertheless, it is quite straightforward that PGA converges asymptotically to critical points (in bounded domains) for differentiable functions. Thus, as long as $\Phi$ is differentiable, it is guaranteed that PGA will asymptotically converge to a critical point of $\Phi$. By Lemma 4.1, this point will be a Nash policy.

## 5    EXPERIMENTS: CONGESTION GAMES

We next study the performance of policy gradient in a general class of MDPs that are congestion games at every state (cf. Roughgarden (2015); Bistritz & Bambos (2020)). As we argued such MDPs are not necessarily MPGs; however, this class of MDPs contains MPGs, e.g., under additional conditions on the transitions, or ordinal (weighted) MPGs under more general conditions.

**Experimental setup.** We consider an experiment (Figure 4) with $N = 8$ agents, $A_i = 4$ facilities (resources or locations) that the agents can select from and $S = 2$ states: a *safe* state and a *distancing* state. In both states, all agents prefer to be in the same facility with as many other agents as possible *(follow the crowd)*. In particular, the reward of each agent for being at facility $k$ is equal to a predefined positive weight $w_k^{\text{safe}}$ times the number of agents at $k = A, B, C, D$. The weights satisfy $w_A^{\text{safe}} < w_B^{\text{safe}} < w_C^{\text{safe}} < w_D^{\text{safe}}$, i.e., facility $D$ is the most preferable by all agents. However, if more than $4 = N/2$ agents find themselves in the same facility, then the game transitions to the distancing state. At the distancing state, the reward structure is the

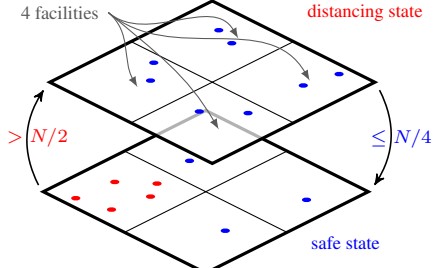

Figure 4: The 2-state MDP.

same for all agents, but reward of each agent is reduced by a constant amount, $c$, where $c > 0$ is a (considerably large) constant. (We also treat the case in which $c$ is different for each facility in Appendix E). To return to the safe state, agents need to achieve maximum distribution over facilities; no more than $2 = N/4$ agents in a facility. We consider deterministic transitions; however, the results are quantitatively equivalent also when these transitions occur with some probability (Appendix E).

**Paremeters.** We perform episodic updates with $T = 20$ steps. At each iteration, we estimate the policy gradients using the average of mini-batches of size 20. We use $\gamma = 0.99$ and a common learning rate $\eta = 0.0001$ (larger than the theoretical guarantee, $\eta = \frac{(1-\gamma)^3}{2\gamma A_{\max}n} \approx 1e - 08$, of Theorem 4.2). Experiments with randomly generated learning rates for each agent, non-deterministic transitions between states and different weights at each facility in the distancing state (which result in non-MPG structure) produce qualitatively equivalent results and are presented in Appendix E.

**Results.** The left panel of Figure 5 shows that the agents learn the expected Nash profile in both states in all runs. Importantly, this (Nash) policy profile is *deterministic* in line with Theorem 4.2. The panels in the middle and right columns depict the L1-accuracy in the policy space at each iteration which is defined as the average distance between the current policy and the final policy of all 8 agents, i.e., L1-accuracy $= \frac{1}{N} \sum_{i \in \mathcal{N}} |\pi_i - \pi_i^{\text{final}}| = \frac{1}{N} \sum_{i \in \mathcal{N}} \sum_s \sum_a |\pi_i(a \mid s) - \pi_i^{\text{final}}(a \mid s)|$.

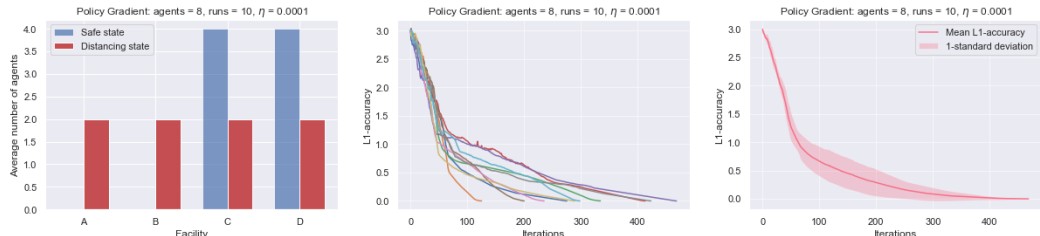

Figure 5: Policy gradient in the 2-state MDP with 8 agents of Section 5. In all runs, the 8 agents learn one of the deterministic Nash policies that leads to the optimal distribution among states (left). Individual trajectories of the L1-accuracy and averages (with 1-standard deviation error bars) show fast convergence in all cases (middle and right columns).

## 6 CONCLUSIONS AND FUTURE DIRECTIONS

We presented positive results about the performance of independent policy gradient in Markov Potential Games (MPGs), a class of multi-agent MDPs that naturally generalize normal-form potential games (games in which agents have aligned incentives) to state-based interactions. We found that MPGs exhibit a richer structure than intuitively expected (e.g., they include MDPs that can be purely competitive at some states), but retain important regularity properties, most importantly, the existence of deterministic Nash policies. In our main result, we showed that independent policy gradient with simultaneous updates converges at a polynomial rate in the approximation error to a Nash policy profile even in the case of finite samples (in that case, using direct parameterization with greedy exploration). MPGs are a first step to cover the gap between the extremes of pure coordination (single-agent or MDPs with identical agents) and pure competition (two-agents, zero-sum MDPs).

**Open Questions.** Our examples and counter-examples of MPGs suggest that multi-agent coordination in state-based interactions is still far from understood and constitutes a major direction for future research in which techniques from cooperative AI (Kao et al., 2021; Mao et al., 2021) or linear MDPs (Jin et al., 2020) may turn out to be particularly useful. From an algorithmic perspective, a second direction for future work involves the analysis of policy gradient with different parameterizations (Agarwal et al., 2020) or of other naturally motivated online learning algorithms (Kleinberg et al., 2009; Panageas et al., 2018; Mehta et al., 2015) and the adaptation of advanced techniques (Cohen et al., 2017; Lee et al., 2019) to reproduce convergence and equilibrium selection results from normal-form to state-based (Markov) games. Alternatively, it would be interesting to test whether complex behavior (e.g., limit cycles, periodic orbits) of learning dynamics that has been observed in restricted normal-form settings Mertikopoulos et al. (2018); Vlatakis-Gkaragkounis et al. (2020) also emerges in the context of exact, weighted, or ordinal MPGs. Finally, turning to efficiency aspects, it would be interesting to measure the losses due to lack of coordination between agents, e.g., via a Price of Anarchy type of analysis Koutsoupias & Papadimitriou (1999); Roughgarden & Tardos (2002).

ACKNOWLEDGMENTS

This research/project is supported in part by the National Research Foundation, Singapore under its AI Singapore Program (AISG Award No: AISG2-RP-2020-016), NRF 2018 Fellowship NRF-NRFF2018-07, NRF2019-NRF-ANR095 ALIAS grant, grant PIE-SGP-AI-2020-01, AME Programmatic Fund (Grant No. A20H6b0151) from the Agency for Science, Technology and Research (A*STAR), Provost's Chair Professorship grant RGEPPV2101 and by EPSRC grant EP/R018472/1.

REPRODUCIBILITY STATEMENT

We included additional technical materials and complete proofs of all the claims and results of the main part in the appendix. We also uploaded the code that was used to run the experiments (policy gradient algorithm) as supplementary material.

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

## A   Additional Notation and Definitions: Section 2

We first provide some additional notation and definitions that will be used in the proofs.

**Q-value and Advantage Functions.**   Recall from the main part that the value function, $V_s^i : \Pi \to \mathbb{R}$, gives the expected reward of agent $i \in \mathcal{N}$ when $s_0 = s$ and the agents draw their actions, $\mathbf{a}_t = (a_{i,t}, \mathbf{a}_{-i,t})$, at time $t \geq 0$ from policies $\pi = (\pi_i, \pi_{-i})$ and is defined as

$$V_s^i(\pi) := \mathbb{E}_{\tau \sim \pi} \left[ \sum_{t=0}^{\infty} \gamma^t r_{i,t} \mid s_0 = s \right].$$

Similarly, we will write $V_\rho^i(\pi) := \mathbb{E}_{s_o \sim \rho}[V_s^i(\pi)]$ to denote the expected value of agent $i \in \mathcal{N}$ under the initial state distribution $\rho$.

For any state $s \in \mathcal{S}$, the Q-value function $Q_s^i : \mathcal{P} \times \mathcal{A} \to \mathbb{R}$ and the advantage function $A_s^i : \mathcal{P} \times \mathcal{A} \to \mathbb{R}$ of agent $i \in \mathcal{N}$ are defined as

$$Q_s^i(\pi, \mathbf{a}) := \mathbb{E}_{\tau \sim \pi} \left[ \sum_{t=0}^{\infty} \gamma^t r_{i,t} \mid s_0 = s, \mathbf{a}_0 = \mathbf{a} \right], \text{ and} \tag{5}$$

$$A_s^i(\pi, \mathbf{a}) := Q_s^i(\pi, \mathbf{a}) - V_s^i(\pi). \tag{6}$$

**Discounted State Distribution.**   It will be useful to define the discounted state visitation distribution $d_{s_0}^\pi(s)$ for $s \in \mathcal{S}$ that is induced by a (joint) policy $\pi$ as

$$d_{s_0}^\pi(s) := (1 - \gamma) \sum_{t=0}^{\infty} \gamma^t \mathrm{Pr}^\pi(s_t = s \mid s_0), \quad \text{for all } s \in \mathcal{S}. \tag{7}$$

As for the value function, we will also write $d_\rho^\pi(s) = \mathbb{E}_{s_0 \sim \rho}[d_{s_0}^\pi(s)]$ to denote the discounted state visitation distribution when the initial state distribution is $\rho$.

## B   Omitted Materials: Section 3

**Proposition B.1** (Separability of Value Functions and Equality of Derivatives). *Let $\mathcal{G} = (\mathcal{S}, \mathcal{N}, \mathcal{A} = \{\mathcal{A}_i\}_{i \in \mathcal{N}}, P, R, \rho)$ be a Markov Potential Game (MPG) with potential $\Phi_s$, for $s \in S$. Then, for the value function $V_s^i, s \in \mathcal{S}$ of each agent $i \in \mathcal{N}$, the following hold*

*P1. Separability of Value Functions: there exists a function $U_s^i : \Delta(\mathcal{A}_{-i})^S \to \mathbb{R}$ such that for each joint policy profile $\pi = (\pi_i, \pi_{-i}) \in \Delta(\mathcal{A})^S$, we have $V_s^i(\pi) = \Phi_s(\pi) + U_s^i(\pi_{-i})$.*

*P2. Equality of Derivatives: the partial derivatives of agent $i$'s value function $V_s^i$ coincide with the partial derivatives of the potential $\Phi_s$ that correspond to agent $i$'s parameters, i.e., $\partial_{x_{i,s,a}} V_s^i(\pi) = \partial_{x_{i,s,a}} \Phi_s(\pi)$, for all $i \in \mathcal{N}$ and all $s \in \mathcal{S}$.*

*Proof.* To obtain P1, consider any 3 arbitrary policies for agent $i$, notated by $\pi_i, \pi_i', \pi_i'' \in \Delta(\mathcal{A}_i)^S$. Then, by the definition of MPGs, we have that

$$\Phi_s(\pi_i, \pi_{-i}) - \Phi_s(\pi_i', \pi_{-i}) = V_s^i(\pi_i, \pi_{-i}) - V_s^i(\pi_i', \pi_{-i}),$$
$$\Phi_s(\pi_i, \pi_{-i}) - \Phi_s(\pi_i'', \pi_{-i}) = V_s^i(\pi_i, \pi_{-i}) - V_s^i(\pi_i'', \pi_{-i}).$$

for every starting state $s \in \mathcal{S}$. This implies that we can write $V_s^i(\pi_i, \pi_{-i})$ as both

$$V_s^i(\pi_i, \pi_{-i}) = \Phi_s(\pi_i, \pi_{-i}) - \Phi_s(\pi_i', \pi_{-i}) + V_s^i(\pi_i', \pi_{-i})$$
$$V_s^i(\pi_i, \pi_{-i}) = \Phi_s(\pi_i, \pi_{-i}) - \Phi_s(\pi_i'', \pi_{-i}) + V_s^i(\pi_i'', \pi_{-i})$$

Thus, we have that $-\Phi_s(\pi_i', \pi_{-i}) + V_s^i(\pi_i', \pi_{-i}) = -\Phi_s(\pi_i'', \pi_{-i}) + V_s^i(\pi_i'', \pi_{-i})$ for any arbitrary pair of policies $\pi_i'$ and $\pi_i''$ for agent $i$, implying that agent $i$'s policy has no impact on these terms. Accordingly, we can express them as

$$U_s^i(\pi_{-i}) := -\Phi_s(\pi_i', \pi_{-i}) + V_s^i(\pi_i', \pi_{-i}) = -\Phi_s(\pi_i'', \pi_{-i}) + V_s^i(\pi_i'', \pi_{-i}),$$

where $U_s^i(\pi_{-i})$ is a function that does not depend on the policy of agent $i$. Thus, we can express the utility function of any agent $i$ in an MPG as

$$V_s^i(\pi) = \Phi_s(\pi) + U_s^i(\pi_{-i}).$$

as claimed. To obtain P2, we use P1 for a vector $x_i$ parameterizing $\pi_i$, and obtain that

$$\partial_{x_{i,a}} V^i(\pi) = \partial_{x_{i,a}} \Phi(\pi) + 0$$

for any coordinate $x_{i,a}$ with $a \in A_i$ of $x_i$, from which we can see that our claim is true. $\square$

Note that P1 serves as a characterization of MPGs. Namely, a multi-agent MDP is an MPG if and only if the value function of each agent $i \in \mathcal{N}$ can be decomposed in a term that is common for all players (potential function) and in a term that may be different for each agent $i \in \mathcal{N}$ but which depends only on the actions of all agents other than $i$. This property carries over from normal form (single state) potential games. Also note that both properties, P1 and P2, hold for any (differentiable for P2) policy parameterization and not only for the direct one that we use here.

*Proof of Theorem 3.1.* Let $\Phi$ be the potential function of $\mathcal{G}$. Since the space $\Delta(\mathcal{A})^S = \Delta(\mathcal{A}_1)^S \times \dots \times \Delta(\mathcal{A}_n)^S$ is compact and $\Phi$ is continuous, $\Phi$ has a global maximum $\Phi_{\max}$. Let $(\pi_1^*, \dots, \pi_n^*)$ denote a global maximizer, i.e., a joint policy profile at which $\Phi_{\max}$ is attained. By the Definitions of MPGs and Nash policies, this implies, in particular, that $(\pi_1^*, \dots, \pi_n^*)$ is a Nash policy, since

$$0 < \Phi_s(\pi_i^*, \pi_{-i}^*) - \Phi_s(\pi_i, \pi_{-i}^*) = V_s^i(\pi_i^*, \pi_{-i}^*) - V_s^i(\pi_i, \pi_{-i}^*), \qquad (*)$$

for all $i \in \mathcal{N}, s \in \mathcal{S}$ and all policies $\pi_i \in \Delta(\mathcal{A}_i)^S$. If $\pi_1^*, \dots, \pi_n^*$ are all deterministic we are done. So, we may assume that there exists an $i \in \mathcal{N}$ so that $\pi_i^*$ is randomized and consider the MDP $\mathcal{G}'$ in which the policy of all agents other than $i$ has been fixed to $\pi_{-i}^*$. $\mathcal{G}'$ is a single agent MDP with the same states as $\mathcal{G}$, the same actions and rewards for agent $i$ and transition probabilities that are determined by the joint distribution of the environment and the joint policy of all agents other than $i$. As a single agent MDP, this setting has a deterministic optimal policy, say $\tilde{\pi}_i$, for agent $i$. Thus, it holds that

$$V_s^i(\tilde{\pi}_i, \pi_{-i}^*) \le V_s^i(\pi_i^*, \pi_{-i}^*) \le V_s^i(\tilde{\pi}_i, \pi_{-i}^*),$$

where the first inequality follows from the fact that $\pi^*$ is a Nash policy and the second from the optimality of $\tilde{\pi}_i$. It follows that

$$V_s^i(\tilde{\pi}_i, \pi_{-i}^*) = V_s^i(\pi_i^*, \pi_{-i}^*),$$

i.e., the payoff of agent $i$ at $(\tilde{\pi}_i, \pi_{-i}^*)$ is the same as in $(\pi_i^*, \pi_{-i}^*)$. Hence, by the definition of the potential function, we have that

$$0 = V_s^i(\tilde{\pi}_i, \pi_{-i}^*) - V_s^i(\pi_i^*, \pi_{-i}^*) = \Phi_s(\tilde{\pi}_i, \pi_{-i}^*) - \Phi_s(\pi_i^*, \pi_{-i}^*),$$

which implies that

$$\Phi_s(\tilde{\pi}_i, \pi_{-i}^*) = \Phi_s(\pi_i^*, \pi_{-i}^*) = \Phi_{\max}.$$

Thus, $(\tilde{\pi}_i, \pi_{-i}^*)$ is also a global maximizer of $\Phi$ which implies that $(\tilde{\pi}_i, \pi_{-i}^*)$ is a Nash policy by the same reasoning as in equation equation $*$. Note that the value of all players other than $i$ may not be the same at the joint policy profile $(\tilde{\pi}_i, \pi_{-i}^*)$ as it is in $(\pi_i^*, \pi_{-i}^*)$. However, what we need for our purpose is that this step reduces the number of randomized policies by one and that it retains the value of the potential function invariant at its global maximum (which ensures that the ensuing policy profile is also a Nash policy). By iterating this process until $\pi_j^*$ becomes deterministic for all agents $j \in \mathcal{N}$, we obtain the claim. $\square$

*Proof of Proposition 3.2.* The proof is constructive and proceeds by finding the potential function, $\Phi_s, s \in \mathcal{S}$ in both cases, C1-C2. Since the individual rewards of the agents at each state $s \in \mathcal{S}$ are captured by a potential function $\phi_s$, then for the reward, $R_i(s, \mathbf{a})$ of each agent $i$ at the action profile $\mathbf{a}$, it holds that

$$R_i(s, \mathbf{a}) = \phi_s(\mathbf{a}) + u_s^i(\mathbf{a}_{-i}), \tag{8}$$

where $u_s^i : \Delta(\mathcal{A}_{-i}) \to \mathbb{R}$ is a function that does not depend on the actions of agent $i$ in any way. Thus, we may write the value function of each agent $i \in \mathcal{N}$ as

$$
\begin{aligned}
V_s^i(\pi) &= \mathbb{E}_{\tau \sim \pi} \left[ \sum\nolimits_{t=0}^{\infty} \gamma^t R_i(s_t, \mathbf{a}_t) \mid s_0 = s \right] \\
&= \mathbb{E}_{\tau \sim \pi} \left[ \sum\nolimits_{t=0}^{\infty} \gamma^t \left( \phi_{s_t}(\mathbf{a}_t) + u_{s_t}^i(\mathbf{a}_{-i,t}) \right) \mid s_0 = s \right] \\
&= \mathbb{E}_{\tau \sim \pi} \left[ \sum\nolimits_{t=0}^{\infty} \gamma^t \phi_{s_t}(\mathbf{a}_t) \mid s_0 = s \right] + \mathbb{E}_{\tau \sim \pi} \left[ \sum\nolimits_{t=0}^{\infty} \gamma^t u_{s_t}^i(\mathbf{a}_{-i,t}) \mid s_0 = s \right] \quad (\star)
\end{aligned}
$$

where $\tau \sim \pi$ is the random trajectory generated by policy $\pi$. To show that $\mathcal{G}$ is an MPG, it suffices to show that the value function of each agent $i \in \mathcal{N}$ can be decomposed in a term that is common for all agents (and which may depend on the actions of agent $i \in \mathcal{N}$) and in a term that does not depend (in any way) in the actions of agent $i$ (dummy term), cf. Proposition B.1. The first term in expression equation $\star$, i.e., $\mathbb{E}_{\tau \sim \pi} \left[ \sum_{t=0}^{\infty} \gamma^t \phi_{s_t}(\mathbf{a}_t) \mid s_0 = s \right]$, depends on the actions of all players and is common for all agents $i \in \mathcal{N}$ (and is thus, a good candidate for the potential function). The second term in expression equation $\star$, i.e., $\mathbb{E}_{\tau \sim \pi} \left[ \sum_{t=0}^{\infty} \gamma^t u_{s_t}^i(\mathbf{a}_{-i,t}) \mid s_0 = s \right]$, does not depend on player $i$ via the payoffs $u_{s_t}^i(\mathbf{a}_{-i,t})$, but, in general, it does depend on player $i$ via the transitions, $\tau \sim \pi$. The two cases in the statement of Proposition 3.2 ensure precisely that this term is independent of the policy of agent $i$, in which case it is a dummy term for agent $i$ or that it is also common for all players and hence, that it can be included in the potential function.

Condition C1. If the transitions do not depend on the action of the players, we have that $\tau \sim P$, where $P$ is an exogenously given distribution function (state-wise). In this case, we have that

$$\Phi_s(\pi) := \mathbb{E}_{\tau \sim \pi} \left[ \sum_{t=0}^{\infty} \gamma^t \phi_{s_t}(\mathbf{a}_t) \mid s_0 = s \right]$$

is a potential function and $U_s^i(\pi_{-i}) := \mathbb{E}_{\tau \sim \pi} \left[ \sum_{t=0}^{\infty} \gamma^t u_{s_t}^i(\mathbf{a}_{-i,t}) \mid s_0 = s \right]$ is a dummy term that does not depend (in any way) on the policy of agent $i \in \mathcal{N}$ which proves the claim.

Condition C2. We will show again that

$$\Phi_{s'}(\pi) := \mathbb{E}_{\tau \sim \pi} \left[ \sum_{t=0}^{\infty} \gamma^t \phi_{s_t}(\mathbf{a}) \mid s_0 = s' \right]$$

is a potential function for $G$ and that the same decomposition as in condition C1 of the value function of agent $i$ in a common and a dummy term applies. To see this, let $\pi_i, \pi_i' \in \Pi_i$ be two policies of agent $i$ and let $\pi = (\pi_i, \pi_{-i}), \pi' = (\pi_i', \pi_{-i})$ where $\pi_{-i}$ is the fixed policy of all agents other than $i$. Then, using equation $\star$, we have that

$$V_{s'}^i(\pi) - V_{s'}^i(\pi') =$$

$$= \Phi_{s'}(\pi) - \Phi_{s'}(\pi') + \mathbb{E}_{\tau \sim \pi} \left[ \sum_{t=0}^{\infty} \gamma^t u_{s_t}^i(\mathbf{a}_{-i,t}) \mid s_0 = s' \right] - \mathbb{E}_{\tau \sim \pi'} \left[ \sum_{t=0}^{\infty} \gamma^t u_{s_t}^i(\mathbf{a}_{-i,t}) \mid s_0 = s' \right].$$

The intermediate value theorem implies that there exists a policy $\xi_i$ which is a convex combination of $\pi_i, \pi_i'$ such that

$$\mathbb{E}_{\tau \sim \pi} \left[ \sum_{t=0}^{\infty} \gamma^t u_{s_t}^i(\mathbf{a}_{-i,t}) \mid s_0 = s' \right] - \mathbb{E}_{\tau \sim \pi'} \left[ \sum_{t=0}^{\infty} \gamma^t u_{s_t}^i(\mathbf{a}_{-i,t}) \mid s_0 = s' \right]$$

$$= (\pi_i - \pi_i')^\top \nabla_{\pi_i} \mathbb{E}_{\tau \sim (\xi_i, \pi_{-i})} \left[ \sum_{t=0}^{\infty} \gamma^t u_{s_t}^i(\mathbf{a}_{-i,t}) \mid s_0 = s' \right]$$

The displayed condition in C2 implies that

$$\nabla_{\pi_i} \mathbb{E}_{\tau \sim (\xi_i, \pi_{-i})} \left[ \sum_{t=0}^{\infty} \gamma^t u_{s_t}^i(\mathbf{a}_{-i,t}) \mid s_0 = s' \right] = (c_s \mathbf{1})_{s \in \mathcal{S}},$$

which is enough to ensure that the dot product in the previous equation is equal to 0 since $\pi_i, \pi_i'$ correspond to probability distributions at every state $s \in \mathcal{S}$, and the sum of their component-wise differences is equal to 0. This proves the claim.

Summing up, in both cases, C1-C2, $\mathcal{G}$ is an MPG as claimed. $\square$

Note that the proof of the (trivial) case in which the instantaneous rewards of all agents $i \in \mathcal{N}$ are equal at each state $s \in \mathcal{S}$ is similar. In this case, it is immediate to see that the instantaneous rewards are precisely given by the potential function at that state, i.e., it holds that $R_i(s, \mathbf{a}) = \phi_s(\mathbf{a})$ for all $i \in \mathcal{N}$ and all $s \in \mathcal{S}$. In this case, it holds that $u_s^i(\mathbf{a}_{-i}) \equiv 0$ for all $i \in \mathcal{N}$ and all $s \in \mathcal{S}$ and hence,

$$\Phi_s(\pi) := \mathbb{E}_{\tau \sim \pi} \left[ \sum_{t=0}^{\infty} \gamma^t \phi_{s_t}(\mathbf{a}) \mid s_0 = s \right]$$

is a potential function for $G$, and the dummy terms are all equal to 0, i.e., $U_s^i(\pi_{-i}) \equiv 0$.

### B.1 EXAMPLES

**Example 1** (Continued). We prove the claim that $\mathcal{G}$ cannot have a deterministic Nash policy profile. To obtain a contradiction, assume that agent $A$ is using a deterministic action $a_A^0 \in \{0, 1\}$ at state 0. Then, agent $B$, who prefers to move to state 1, will optimize their utility by choosing the action $a_B^0 \in \{0, 1\}$ that yields $a_A^0 \oplus a_B^0 = 1$. In other words, given any deterministic action of agent $A$ at state 0, agent $B$ can choose an action that always moves the sequence of play to state 1. Thus, such an action cannot be optimal for agent $A$ which implies that the MDP does not have a deterministic optimal policy profile as claimed.

**Example 2** (Continued). At each state, $s \in \{0, 1\}$, the agents' payoffs, $(R_s^1, R_s^2)$, form a potential game (at that state), and are given as follows

$$\text{State } 0: \quad (R_0^1, R_0^2) = \begin{array}{c} \\ 0 \\ 1 \end{array} \begin{array}{c} 0 \qquad 1 \\ \begin{pmatrix} 5, \ \ 2 & -1, -2 \\ -5, -4 & 1, \ \ 4 \end{pmatrix} \end{array}, \quad \text{with potential} \ \ \Phi_0 = \begin{pmatrix} 4 & 0 \\ -6 & 2 \end{pmatrix},$$

$$\text{State } 1: \quad (R_1^1, R_1^2) = (0, 0), \quad \text{with potential} \ \ \Phi_1 = 0.$$

In this MDP, agents need only to select an action at state $s_0$. Thus, we will denote a policy, $\pi_1$, of agent 1 by $\pi_1 = (p, 1 - p)$ where $p \in [0, 1]$ is the probability with which agent $A$ selects action 0 at state $s_0$. Similarly, we will denote a policy, $\pi_2$, of agent $B$ by $\pi_2 = (q, 1 - q)$ where $q \in [0, 1]$ is the probability with which agent $B$ selects action 0 at state $s_0$. Moreover, we will slightly abuse notation and write

$$R_0^i(\pi) = R_0^i(\pi_1, \pi_2) = \pi_1^\top R_0^i \pi_2 = [p, 1 - p] R_0^i [q, 1 - q]^\top.$$

We also assume that the horizon is infinite and there is a discount factor $\gamma \in [0, 1)$. Accordingly, we can calculate the value functions $V_0^i(\pi_1, \pi_2)$ of agents $i = A, B$ starting from state $s_0$ as follows,

$$V_0^i(\pi) = R_0^i(\pi) + \gamma pq V_0^i(\pi) - \gamma(1 - pq) \times 0$$

which yields the solution

$$V_0^i(\pi) = \frac{R_0^i(\pi)}{1 - \gamma pq}, \quad \text{for } i = A, B.$$

Next, we use the Performance Difference Lemma (Lemma 3.2 by Agarwal et al. (2020)) to determine the difference in the value between two different policies. We will do this for agent 1 (the calculation

is similar for agent 2: we use here 1 for agent 1 and 2 for agent $B$). For a policy $\pi = (\pi_1, \pi_2)$, we have at state $s_0$ that

$$\mathbb{E}_{a \sim \pi_1(\cdot|s_0)}[A_0^1(\pi', \mathbf{a})] = p A_0^1(\pi', 0, a_2) + (1-p) A^1(\pi', 1, a_2)$$

$$= p \left[ R_0^1(0, \pi_2) + \gamma q V_0(\pi') - V_0(\pi') \right] + (1-p) \left[ R_0^1(1, \pi_2) + 0 - V_0(\pi') \right]$$

$$= p R_0^1(0, \pi_2) + (1-p) R_0^1(0, \pi_2) - (1 - \gamma pq) V_0(\pi')$$

$$= R_0^1(\pi) - (1 - \gamma pq) V_0(\pi').$$

At state $s_1$, there is only one available action for each agent which yields a payoff of 0. Thus,

$$\mathbb{E}_{a \sim \pi_1(\cdot|s_1)}[A_1^1(\pi', \mathbf{a})] = 0.$$

Moreover, concerning the discounted visitation distribution, we have that

$$d_0^\pi(s_0) = (1-\gamma) \sum_{t=0}^{\infty} \gamma^t \mathrm{Pr}^\pi(s_t = s_0 \mid s_0) = (1-\gamma) \left[ 1 + \gamma pq + (\gamma pq)^2 + \dots \right] = \frac{1-\gamma}{1 - \gamma pq},$$

and $d_0^\pi(s_1) = 1 - d_0^\pi(s_0) = \frac{\gamma(1 - pq)}{1 - \gamma pq}$. Thus, using all the above, we have that

$$V_0(\pi) - V_0(\pi') = \frac{1}{1-\gamma} \left[ \frac{1-\gamma}{1 - \gamma pq} \cdot (R_0^1(\pi) - (1 - \gamma pq) V_0(\pi')) + \frac{\gamma(1-pq)}{1 - \gamma pq} \cdot 0 \right]$$

$$= \frac{R_0^1(\pi)}{1 - \gamma pq} - V_0(\pi') = V_0(\pi) - V_0(\pi').$$

which shows that our initial calculations conform with the outcome specified by the Performance Difference Lemma.

Finally, a direct calculation shows that $\Phi_s = \phi_s$ for $s = 0, 1$ is a valid potential function for which the MDP is an *ordinal* MPG.

**Example 3** (Continued). At state $s_0$, we consider the game with action sets $A_1(s_0) = A_2(s_0) = \{H, T\}$ and (instantaneous) payoffs

$$R_1(s_0, a_1, a_2) = \begin{array}{c} \\ H \\ T \end{array} \begin{array}{cc} H & T \\ \begin{pmatrix} 1 & -1 \\ -1 & 1 \end{pmatrix} \end{array} \quad \text{and} \quad R_2(s_0, a_1, a_2) = \begin{array}{c} \\ H \\ T \end{array} \begin{array}{cc} H & T \\ \begin{pmatrix} -1 & 1 \\ 1 & -1 \end{pmatrix} \end{array},$$

where $a_1$ denotes the action of agent 1 and $a_2$ the action of agent 2 (agent 1 selects rows and agent 2 selects columns in both matrices). This is a constant sum game (equivalent to zero-sum) and hence, it is not an (ordinal) potential game. Apart from the instantaneous rewards, agents' actions at $s_0$ induce a deterministic transition to a state in which the only available actions to the agents are precisely the actions that they chose at state $s_0$ and their instantaneous rewards at this state are the rewards of the other agent at $s_0$. In particular, there are four possible transitions to states $s_{ab}$ with $a, b \in \{H, T\}$, with action sets and instantaneous rewards given by

$$A_1(s_{ab}) = \{a\}, A_2(s_{ab}) = \{b\}, \quad R_1(s_{ab}, a, b) = R_2(s_0, b, a), \quad R_2(s_{ab}, a, b) = R_1(s_0, b, a),$$

for agents 1 and 2, respectively. Note that the visitation probability of this states is equal to the visitation probability of state $s_0$. After visiting one of these states, the MDP transitions to state $s_1$ which is a potential game, with potential function given by

$$\Phi_1 = \begin{array}{c} \\ L \\ R \end{array} \begin{array}{cc} L & R \\ \begin{pmatrix} 4 & 3 \\ 3 & 0 \end{pmatrix} \end{array}$$

As mentioned above, the game in state $s_0$ does not admit a potential function. However, the joined rewards $RJ_1, RJ_2$ of agents 1 and 2 which result from selecting an action profile $(a, b) \in H, T^2$ at $s_0$ and then traversing both $s_0$ and the ensuing $s_{ab}$ (part included in the dotted rectangle in Figure 3), do admit a potential function. The potential function in this case is the sum of agents' rewards and is given by

$$\Phi_{0ab} = \begin{array}{c} \\ H \\ T \end{array} \begin{array}{cc} H & T \\ \begin{pmatrix} 1-1 & 1-1 \\ 1-1 & 1-1 \end{pmatrix} \end{array} = \begin{array}{c} \\ H \\ T \end{array} \begin{array}{cc} H & T \\ \begin{pmatrix} 0 & 0 \\ 0 & 0 \end{pmatrix} \end{array}.$$

Let $\pi_1 = (\mathbf{x}_0, \mathbf{x}_1)$ denote a policy of agent 1. Here $\mathbf{x}_0 = (x_0, 1 - x_0)$, where $x_0 \in [0, 1]$ is the probability with which agent 1 chooses action $H$ at state $s_0$. Similarly, $\mathbf{x}_1 = (x_1, 1 - x_1)$ where $x_1 \in [0, 1]$ is the probability with which agent 1 chooses action $L$ at state $s_1$. At states $s_{ab}, a, b \in \{H, T\}^2$, agents only have one action to choose from, so this choice is eliminated from their policy representation. Similarly, we represent a policy of agent 2 by $\pi_2 = (\mathbf{y}_0, \mathbf{y}_1)$ with $y_0, y_1 \in [0, 1]$. Let also

$$p_0 := p_0(\pi_1, \pi_2) := \Pr(s_{t+1} = s_0 \mid s_t = s_1, \pi_1, \pi_2), \tag{9}$$

In the general case, $p_0$, i.e., the transition probability from $s_1$ to $s_0$, may depend on the actions of the agents or it may be completely exogenous (i.e., constant with respect to agents' actions). If we write

$$p_0(a_1, a_2) := \Pr(s_{t+1} = s_0 \mid s_t = s_1, a_1, a_2), \quad \text{for } a_1, a_2 \in \{L, R\},$$

to denote the probability of transitioning from state $s_1$ to state $s_0$ given that the agents chose actions $a_1, a_2 \in \{L, R\}$ at state $s_1$, then we can write $p_0$ as

$$p_0 = \mathbb{E}_{(a_1, a_2) \sim (\pi_1, \pi_2)}[p_0(a_1, a_2)] = \sum_{(a_1, a_2) \in \{L, R\}^2} \Pr(a_1, a_2 \mid \pi_1, \pi_2) \cdot p_0(a_1, a_2)$$

$$= x_1 y_1 p_0(L, L) + x_1(1 - y_1) p_0(L, R) + (1 - x_1) y_1 p_0(R, L) + (1 - x_1)(1 - y_1) p_0(R, R). \tag{10}$$

Using this notation, we can now proceed to compute the value function of each state of the MDP in Figure 3. Since the value of states $s_{a,b}, a, b \in H, T^2$ is equal to a constant reward plus the value of state $s_1$ (discounted by $\gamma$), it suffices to calculate the value for states $s_0$ and $s_1$. We have that

$$V_0^1(\pi_1, \pi_2) = \mathbf{x}_0 R_0^1 \mathbf{y}_0 + \gamma \left( \mathbf{x}_0 \left( R_0^2 / \gamma \right) \mathbf{y}_0 \right) + \gamma^2 V_1^1(\pi_1, \pi_2)$$
$$V_1^1(\pi_1, \pi_2) = \mathbf{x}_1 R_1^1 \mathbf{y}_1 + \gamma \left[ p_0 V_0^1(\pi_1, \pi_2) + (1 - p_0) V_1^1(\pi_1, \pi_2) \right],$$

which after some trivial calculations yield

$$V_0^1(\pi_1, \pi_2) = \mathbf{x}_0 \left( R_0^1 + R_0^1 \right) \mathbf{y}_0 + \gamma^2 V_1^1(\pi_1, \pi_2)$$
$$V_1^1(\pi_1, \pi_2) = \frac{1}{1 - \gamma(1 - p_0)} \left[ \mathbf{x}_1 R_1^1 \mathbf{y}_1 + \gamma p_0 V_0^1(\pi_1, \pi_2) \right].$$

This is a system of 2 equations in the 2 unknown quantities, $V_0^1(\pi_1, \pi_2)$ and $V_1^1(\pi_1, \pi_2)$. Solving for these two quantities, yields the unique solution

$$V_0^1(\pi_1, \pi_2) = \frac{1}{1 - \gamma(1 - p_0) - \gamma^3 p_0} \left[ (1 - \gamma(1 - p_0)) \mathbf{x}_0 \left( R_0^1 + R_0^2 \right) \mathbf{y}_0 + \gamma^2 \mathbf{x}_1 R_1^1 \mathbf{y}_1 \right].$$
$$V_1^1(\pi_1, \pi_2) = \frac{1}{1 - \gamma(1 - p_0) - \gamma^3 p_0} \left[ \gamma p_0 \mathbf{x}_0 \left( R_0^1 + R_0^2 \right) \mathbf{y}_0 + \mathbf{x}_1 R_1^1 \mathbf{y}_1 \right].$$

In the case that $p_0$ is a constant with respect to $\pi_1, \pi_2$, then both value functions are of the form

$$V_s^i(\pi_1, \pi_2) = c_1(s) \cdot \mathbf{x}_0 \left( R_0^1 + R_0^2 \right) \mathbf{y}_0 + c_2(s) \cdot \mathbf{x}_1 R_1^i \mathbf{y}_1, \quad \text{for } s \in \{s_0, s_1\}, \text{ and } i = \{1, 2\},$$

where $c_1(s), c_2(s) > 0$ are appropriate constants that depend only the state $s \in \{s_0, s_1\}$ and on agents 1, 2. Since the game at $s_1$ is a potential game, with potential function given by a $2 \times 2$ matrix $\Phi_1$, it is immediate to infer that

$$V_s^i(\pi_1', \pi_2) - V_s^i(\pi_1, \pi_2) = \Phi_s(\pi_1', \pi_2) - \Phi_s(\pi_1, \pi_2), \text{ for } s \in \{s_0, s_1\},$$

with

$$\Phi_s(\pi_1, \pi_2) := c_1(s) \mathbf{x}_0 \left( R_0^1 + R_0^2 \right) \mathbf{y}_0 + c_2(s) \cdot \mathbf{x}_1 \Phi_1 \mathbf{y}_1, \text{ for } s \in \{s_0, s_1\}.$$

However, if $p_0$ depends on the actual policies of agents 1 and 2, cf. equation equation 9, then it is not immediate to determine a potential (or even to decide whether a (exact) potential exists or not).

*Remark* 2. Several elements of Figure 3 have been selected in the sake of simplicity and are not necessary for the main takeaway, i.e., that there are MDP that are not potential at some states, but which are MPGs. First, the transitions from $s_0$ to the states $s_{ab}$ need not be deterministic. To see this, let $q \in (0, 1)$ and assume that if the agents select actions $H, T$ is $s_0$, then the process transitions with probability $q$ to a state $s_{HT}$ with rewards $(-1, 1)/q\gamma$ and with probability $(1 - q)$ to a state $s'_{HT}$

with rewards $(1, -1)/(1 - q)\gamma$. The rest remains the same. Accordingly, the expected reward for agent 1 after $(H, T)$ has been selected in $s_0$ is the same as in the current format.

Second, the construction with states $s_0$ and $s_{ab}, (a, b) \in H, T^2$ is not the only one that leads to such an example. Another very common instance occurs in the case of *aliasing* between $s_0$ and states $s_{ab}$, i.e., when the agents cannot tell these states apart. The intuition which carries over from the currently presented example is that the roles of the agents are essentially reversed between the two states, but the agents do not know (from the observable features) in which state they are. Thus, any valid policy, selects the same action in both states leading to the same situation as in the presented example.

Finally, if the horizon is finite, then the instantaneous rewards in states $s_{ab}$ still work if we eliminate the scaling factor (here $\gamma$). Thus, the construction works in both episodic and continuing settings.

## C  AUXILIARY LEMMAS

Before proceeding to the omitted materials of Section 4, we first include some auxiliary results that we will use in the proofs presented there.

Recall that $P_\mathcal{X}$ denotes the projection onto some set $\mathcal{X}$.

**Lemma C.1** (Bubeck (2015), Lemma 3.6). *Let $f$ be a $\beta$-smooth function[5] with convex domain $\mathcal{X}$. Let $x \in \mathcal{X}$, $x^+ = P_\mathcal{X}(x - \frac{1}{\beta}\nabla f(x))$ and $g_\mathcal{X}(x) = \beta(x - x^+)$. Then the following holds true:*

$$f(x^+) - f(x) \leq -\frac{1}{2\beta} \|g_\mathcal{X}(x)\|_2^2.$$

**Lemma C.2** (Agarwal et al. (2020), Proposition B.1). *Let $f(\pi)$ be a $\beta$-smooth function in $\pi \in \Delta(\mathcal{A})^S$. Define the gradient mapping*

$$G(\pi) = \beta \left( P_{\Delta(\mathcal{A})^S} \left( \pi + \frac{1}{\beta}\nabla_\pi f(\pi) \right) - \pi \right)$$

*and the update rule for the projected gradient is $\pi' = \pi + \frac{1}{\beta}G(\pi)$. If $\|G(\pi)\|_2 \leq \epsilon$, then*

$$\max_{\pi + \delta \in \Delta(\mathcal{A})^S, \|\delta\|_2 \leq 1} \delta^\top \nabla_\pi f(\pi') \leq 2\epsilon.$$

**Lemma C.3.** *Let $\Phi_\mu(\pi)$ be the potential function (which is $\beta$-smooth) and assume $\pi \in \Delta(\mathcal{A})^S$ uses $\alpha$-greedy parameterization. Define the gradient mapping*

$$G(\pi) = \beta \left( P_{\Delta(\mathcal{A})^S} \left( \pi + \frac{1}{\beta}\nabla\Phi_\mu(\pi) \right) - \pi \right)$$

*and the update rule for the projected gradient is $\pi' = \pi + \frac{1}{\beta}G(\pi)$. If $\|G(\pi)\|_2 \leq \epsilon$, then*

$$\max_{\pi + \delta \in \Delta(\mathcal{A})^S, \|\delta\|_2 \leq 1} \delta^\top \nabla\Phi_\mu(\pi') \leq 2\epsilon + \sqrt{n\alpha A_{\max}}.$$

*Proof.* It is a direct application of Lemma C.2 and the fact that $\Phi_\mu$ is Lipschitz with parameter $\frac{\sqrt{nA_{\max}}}{(1-\gamma)^2}$ (this is Proposition 3 in page 23 of Daskalakis et al. (2020)). $\square$

**Lemma C.4** (Unbiased with bounded variance Daskalakis et al. (2020)). *It holds that $\hat{\nabla}_{\pi_i}^{(t)}$ is unbiased estimator of $\nabla_{\pi_i} V^i$ for all $i$, that is $\mathbb{E}_{\pi^{(t)}} \hat{\nabla}_{\pi_i}^{(t)} = \nabla_{\pi_i} V_\rho^i(\pi^{(t)})$ for all $i$. Moreover, for all agents $i$ we get that (this is what the authors actually prove), it holds that $\mathbb{E}_{\pi^{(t)}} \left\| \hat{\nabla}_{\pi_i}^{(t)} \right\|_2^2 \leq \frac{24A_{\max}^2}{\alpha(1-\gamma)^4}$.*

## D  OMITTED MATERIALS: SECTION 4

Before we proceed with the formal statements and proofs of this section, we recall the commonly used definition of distribution mismatch coefficient Kakade & Langford (2002) applied to our setting.

**Definition 5** (Distribution Mismatch coefficient). For any distribution $\mu \in \Delta(\mathcal{S})$ and any policy $\pi \in \Pi$, we call $D := \max_{\pi \in \Pi} \left\| \frac{d_\mu^\pi}{\mu} \right\|_\infty$ the *distribution mismatch coefficient*, where $d_\mu^\pi$ is the discounted state distribution (7).

---

[5]Differentiable with $\nabla f$ to be $\beta$-Lipschitz.

## D.1 EXACT GRADIENTS CASE

The first auxiliary Lemma has to do with the projection operator that is used on top of the independent policy gradient, so that the policy vector $\pi_i^{(t)}$ remains a probability distribution for all agents $i \in \mathcal{N}$ (see (PGA)). It is not hard to show (due to separability) that the projection operator being applied independently for each agent $i$ on $\Delta(\mathcal{A}_i)^S$ is the same as jointly applying projection on $\Delta(\mathcal{A})^S$. This is the statement of Lemma D.1.

**Lemma D.1** (Projection Operator). *Let $\pi := (\pi_1, ..., \pi_n)$ be the policy profile for all agents and let $\pi' = \pi + \eta \nabla_\pi \Phi_\rho(\pi)$ be a gradient step on the potential function for a step-size $\alpha > 0$. Then,*

$$P_{\Delta(\mathcal{A})^S}(\pi') = (P_{\Delta(\mathcal{A}_1)^S}(\pi'_1), \ldots, P_{\Delta(\mathcal{A}_n)^S}(\pi'_n)).$$

*Proof of Lemma D.1.* Observe that $P_{\mathcal{X}}(y) = \arg\min_{x \in \mathcal{X}} \|x - y\|_2^2$ for any set $\mathcal{X} \subseteq \mathbb{R}^n$. Thus,

$$P_{\Delta(\mathcal{A})^S}(y) = \arg\min_{x \in \Delta(\mathcal{A})^S} \|x - \pi'\|_2^2 = \arg\min_{x_1 \in \Delta(\mathcal{A}_1)^S, \ldots, x_n \in \Delta(\mathcal{A}_n)^S} \sum_{i=1}^n \|x_i - \pi'_i\|_2^2$$

$$= \sum_{i=1}^n \arg\min_{x_i \in \Delta(\mathcal{A}_i)^S} \|x_i - \pi'_i\|_2^2 = (P_{\Delta(\mathcal{A}_1)^S}(\pi'_1), \ldots, P_{\Delta(\mathcal{A}_n)^S}(\pi'_n)). \qquad \square$$

The main implication of Lemma D.1 along with the equality of the derivatives between value functions and the potential function in MPGs, i.e., $\nabla_{\pi_i} V_s^i(\pi) = \nabla_{\pi_i} \Phi(\pi)$ for all $i \in \mathcal{N}$ (see property P2 in Proposition B.1), is that running independent equation PGA on each agent's value function is equivalent to running equation PGA on the potential function $\Phi$. In turn, Lemma 4.1 suggests that as long as policy gradient reaches a point $\pi^{(t)}$ with small gradient along the directions in $\Delta(\mathcal{A})^S$, it must be the case that $\pi^{(t)}$ is an approximate Nash policy. Together with Lemma D.1, this will be sufficient to prove convergence of equation PGA.

*Proof of Lemma 4.1.* Fix agent $i$ and suppose that $i$ deviates to an optimal policy $\pi_i^*$ (w.r.t the corresponding single agent MDP). Since $\pi$ is $\epsilon$-stationary it holds that (Definition 4)

$$\max_{\pi'_i \in \Delta(\mathcal{A}_i)^S} (\pi'_i - \pi_i)^\top \nabla_{\pi_i} \Phi_\mu(\pi) \le \sqrt{S}\epsilon. \tag{11}$$

Thus, with $\pi^* = (\pi_i^*, \pi_{-i})$, Lemma D.3 implies that

$$\Phi_\rho(\pi^*) - \Phi_\rho(\pi) \le \frac{1}{1-\gamma} \left\| \frac{d_\rho^{\pi^*}}{\mu} \right\|_\infty \max_{\pi' = (\pi'_i, \pi_{-i})} (\pi' - \pi)^\top \nabla_{\pi_i} \Phi_\mu(\pi) \overset{(11)}{\le} \frac{D}{1-\gamma} \sqrt{S}\epsilon. \tag{12}$$

Thus, using the definition of the potential function (cf. Definition 2), we obtain that

$$V_\rho^i(\pi^*) - V_\rho^i(\pi) = \Phi_\rho(\pi^*) - \Phi_\rho(\pi) \le \frac{\sqrt{S}D\epsilon}{1-\gamma}.$$

Since the choice of $i$ was arbitrary, we conclude that $\pi$ is an $\frac{\sqrt{S}D\epsilon}{1-\gamma}$-approximate Nash policy. $\qquad \square$

To prove Lemma D.3, we will use a multi-agent version of the Performance Difference Lemma (cf. Agarwal et al. (2020) for a single agent and Daskalakis et al. (2020) for two agents).

**Lemma D.2** (Multi-agent Performance Difference Lemma). *Consider an $n$-agent MDP $\mathcal{G}$ and fix an agent $i \in \mathcal{N}$. Then, for any policies $\pi = (\pi_i, \pi_{-i}), \pi' = (\pi'_i, \pi_{-i}) \in \Pi$ and any distribution $\rho \in \Delta(S)$, it holds that*

$$V_\rho^i(\pi) - V_\rho^i(\pi') = \frac{1}{1-\gamma} \mathbb{E}_{s \sim d_\rho^\pi} \mathbb{E}_{a_i \sim \pi_i(\cdot|s)} \mathbb{E}_{\mathbf{a}_{-i} \sim \pi_{-i}(\cdot|s)} \left[ A_s^i(\pi', a_i, \mathbf{a}_{-i}) \right],$$

*where $\mathbf{a}_{-i} \sim \pi_{-i}(\cdot \mid s)$ denotes the action profile of all agents other than $i$ that is drawn from the product distribution induced by their policies $\pi_{-i} = (\pi_j)_{j \ne i \in \mathcal{N}} \in \Pi_{-i}$.*

*Proof.* For any initial state $s \in \mathcal{S}$ and joint policies $\pi = (\pi_i, \pi_{-i}), \pi' = (\pi'_i, \pi_{-i}) \in \Pi$, it holds that

$$
\begin{aligned}
V_s^i(\pi) - V_s^i(\pi') &= \mathbb{E}_{\tau \sim \pi} \left[ \sum_{t=0}^{\infty} \gamma^t r_{i,t} \mid s_0 = s \right] - V_s^i(\pi') \\
&= \mathbb{E}_{\tau \sim \pi} \left[ \sum_{t=0}^{\infty} \gamma^t \left( r_{i,t} - V_{s_t}(\pi') + V_{s_t}(\pi') \right) \mid s_0 = s \right] - V_s^i(\pi') \\
&= \mathbb{E}_{\tau \sim \pi} \left[ \sum_{t=0}^{\infty} \gamma^t \left( r_{i,t} - V_{s_t}(\pi') + \gamma V_{s_{t+1}}(\pi') \right) \mid s_0 = s \right] \\
&= \mathbb{E}_{\tau \sim \pi} \left[ \sum_{t=0}^{\infty} \gamma^t \left( r_{i,t} + \gamma \mathbb{E} \left[ V_{s_{t+1}}(\pi') \mid s_t, a_{i,t}, \mathbf{a}_{-i,t} \right] - V_{s_t}(\pi') \right) \mid s_0 = s \right] \\
&= \mathbb{E}_{\tau \sim \pi} \left[ \sum_{t=0}^{\infty} \gamma^t A_{s_t}^i(\pi', \mathbf{a}_t) \mid s_0 = s \right] \\
&= \frac{1}{1-\gamma} \mathbb{E}_{s' \sim d_\rho^\pi} \mathbb{E}_{a_i \sim \pi_i(\cdot \mid s')} \mathbb{E}_{\mathbf{a}_{-i} \sim \pi_{-i}(\cdot \mid s')} \left[ A_{s'}^i(\pi', \mathbf{a}) \right].
\end{aligned}
$$

Taking expectation over the states $s \in \mathcal{S}$ with respect to the distribution $\rho \in \Delta(\mathcal{S})$ yields the result. $\qquad\square$

To prove Lemma 4.1, we will need an agent-wise version of the gradient domination property of single-agent MDPs (cf. Lemma 4.1 in Agarwal et al. (2020)). This is presented in Lemma D.3.

**Lemma D.3** (Agent-wise Gradient Domination Property in MPGs). *Let $\mathcal{G}$ be an MPG with potential function $\Phi$, fix any agent $i \in \mathcal{N}$, and let $\pi = (\pi_i, \pi_{-i}) \in \Delta(\mathcal{A})^S$ be a policy. Let $\pi_i^*$ be an optimal policy for agent $i$ in the single agent MDP in which the rest of the agents are fixed to choose $\pi_{-i}$. Then, for the policy $\pi^* = (\pi_i^*, \pi_{-i}) \in \Delta(\mathcal{A})^S$ that differs from $\pi$ only in the policy component of agent $i$, and for any distributions $\mu, \rho \in \Delta(\mathcal{S})$, it holds that*

$$
\Phi_\rho(\pi^*) - \Phi_\rho(\pi) \leq \frac{1}{1-\gamma} \left\| \frac{d_\rho^{\pi^*}}{\mu} \right\|_\infty \max_{\pi' = (\pi'_i, \pi_{-i})} (\pi' - \pi)^\top \nabla_{\pi_i} \Phi_\mu(\pi),
$$

*Proof of Lemma D.3.* Fix an agent $i \in \mathcal{N}$ and let $\pi = (\pi_i, \pi_{-i}), \pi^* = (\pi_i^*, \pi_{-i}) \in \Pi = \Delta(\mathcal{A})^S$. By the definition of MPGs (cf. Definition 2), it holds that

$$
V_\rho^i(\pi^*) - V_\rho^i(\pi) = \Phi_\rho(\pi^*) - \Phi_\rho(\pi).
$$

Thus, using the multi-agent version of the Performance Difference Lemma (cf. Lemma D.2), we have for any distribution $\mu \in \Delta(\mathcal{S})$ that

$$
\begin{aligned}
\Phi_\rho(\pi^*) - \Phi_\rho(\pi) &= V_\rho^i(\pi^*) - V_\rho^i(\pi) \\
&= \frac{1}{1-\gamma} \mathbb{E}_{s \sim d_\rho^{\pi^*}} \mathbb{E}_{a_i \sim \pi_i^*(\cdot \mid s)} \mathbb{E}_{\mathbf{a}_{-i} \sim \pi_{-i}^*(\cdot \mid s)} \left[ A_s^i(\pi, a_{i,t}, \mathbf{a}_{-i,t}) \right] \\
&\leq \frac{1}{1-\gamma} \max_{\pi'_i \in \Pi_i} \left\{ \sum_{s \in \mathcal{S}} d_\rho^{\pi^*}(s) \mathbb{E}_{a_i \sim \pi'_i(\cdot \mid s)} \mathbb{E}_{\mathbf{a}_{-i} \sim \pi_{-i}^*(\cdot \mid s)} \left[ A_s^i(\pi, a_i, \mathbf{a}_{-i}) \right] \right\}, \\
&= \frac{1}{1-\gamma} \max_{\pi'_i \in \Pi_i} \left\{ \sum_{s \in \mathcal{S}} \frac{d_\rho^{\pi^*}(s)}{d_\mu^\pi(s)} d_\mu^\pi(s) \mathbb{E}_{a_i \sim \pi'_i(\cdot \mid s)} \mathbb{E}_{\mathbf{a}_{-i} \sim \pi_{-i}^*(\cdot \mid s)} \left[ A_s^i(\pi, a_i, \mathbf{a}_{-i}) \right] \right\}, \\
&\leq \frac{1}{1-\gamma} \left\| \frac{d_\rho^{\pi^*}}{d_\mu^\pi} \right\|_\infty \max_{\pi'_i \in \Pi_i} \left\{ \sum_{s \in \mathcal{S}} d_\mu^\pi(s) \mathbb{E}_{a_i \sim \pi'_i(\cdot \mid s)} \mathbb{E}_{\mathbf{a}_{-i} \sim \pi_{-i}^*(\cdot \mid s)} \left[ A_s^i(\pi, a_i, \mathbf{a}_{-i}) \right] \right\}.
\end{aligned}
$$

To proceed, observe that

$$
\mathbb{E}_{a_i \sim \pi_i(\cdot \mid s)} \mathbb{E}_{\mathbf{a}_{-i} \sim \pi_{-i}^*(\cdot \mid s)} \left[ A_s^i(\pi, a_i, \mathbf{a}_{-i}) \right] = 0.
$$

Thus, for any $\pi'_i \in \Pi_i$ and any state $s \in \mathcal{S}$, it holds that

$$
\begin{aligned}
\mathbb{E}_{a_i \sim \pi'_i(\cdot \mid s)} \mathbb{E}_{\mathbf{a}_{-i} \sim \pi_{-i}^*(\cdot \mid s)} & \left[ A_s^i(\pi, a_i, \mathbf{a}_{-i}) \right] = \\
&= \sum_{a_i \in \mathcal{A}_i} \left( \pi'_i(a_i \mid s) - \pi_i(a_i \mid s) \right) \mathbb{E}_{\mathbf{a}_{-i} \sim \pi_{-i}^*(\cdot \mid s)} \left[ A_s^i(\pi, a_i, \mathbf{a}_{-i}) \right] \\
&= \sum_{a_i \in \mathcal{A}_i} \left( \pi'_i(a_i \mid s) - \pi_i(a_i \mid s) \right) \mathbb{E}_{\mathbf{a}_{-i} \sim \pi_{-i}^*(\cdot \mid s)} \left[ Q_s^i(\pi, a_i, \mathbf{a}_{-i}) \right]
\end{aligned}
$$

since $V_s^i(\pi)$ does not depend on $a_i$. Substituting back in the last inequality of the previous calculations, we obtain that

$$\Phi_\rho(\pi^*) - \Phi_\rho(\pi) \leq$$
$$\leq \left\| \frac{d_\rho^{\pi^*}}{d_\mu^\pi} \right\|_\infty \max_{\pi_i' \in \Pi_i} \left\{ \sum_{s,a_i} \frac{d_\mu^\pi(s)}{1 - \gamma} \left( \pi_i'(a_i \mid s) - \pi_i(a_i \mid s) \right) \mathbb{E}_{\mathbf{a}_{-i} \sim \pi_{-i}^*(\cdot|s)} \left[ Q_s^i(\pi, a_i, \mathbf{a}_{-i}) \right] \right\}$$
$$= \left\| \frac{d_\rho^{\pi^*}}{d_\mu^\pi} \right\|_\infty \max_{\pi_i' \in \Pi_i} (\pi_i' - \pi_i)^\top \nabla_{\pi_i} V_\mu^i(\pi),$$

where we used the policy gradient theorem (Agarwal et al. (2020); Sutton & Barto (2018)) under the assumption of direct policy parameterization. We can further upper bound the last expression by using that $d_\mu^\pi(s) \geq (1 - \gamma)\mu(s)$ which follows immediately from the definition of the discounted visitation distribution $d_\mu^\pi(s)$ for any initial state distribution $\mu$. Finally, property P2 of Proposition B.1, implies that $\nabla_{\pi_i} V_\rho^i(\pi) = \nabla_{\pi_i} \Phi_\rho(\pi)$ (making crucial use of the MPG structure). Putting these together, we have that

$$\Phi_\rho(\pi^*) - \Phi_\rho(\pi) \leq \frac{1}{1 - \gamma} \left\| \frac{d_\rho^{\pi^*}}{\mu} \right\|_\infty \max_{\pi' = (\pi_i', \pi_{-i}^*)} (\pi' - \pi)^\top \nabla_{\pi_i} V_\mu^i(\pi)$$
$$= \frac{1}{1 - \gamma} \left\| \frac{d_\rho^{\pi^*}}{\mu} \right\|_\infty \max_{\pi' = (\pi_i', \pi_{-i}^*)} (\pi' - \pi)^\top \nabla_{\pi_i} \Phi_\mu(\pi),$$

as claimed. $\qquad \square$

*Remark* 3. Intuitively, Lemma D.3 implies that there is a *best response* structure in the agents' updates that we can exploit to show convergence of (projected) policy gradient to a Nash policy profile. In particular, given a fixed policy profile of all agents other than $i$, the decision of agent $i$ is equivalent to the decision of that agent in a single MDP. Thus, the following inequality

$$V_s^i(\pi^*) - V_s^i(\pi) \leq \frac{1}{1 - \gamma} \left\| \frac{d_s^{\pi^*}}{\mu} \right\|_\infty \max_{\pi' = (\pi_i', \pi_{-i})} (\pi' - \pi)^\top \nabla_{\pi_i} V_\mu^i(\pi)$$

(which stems directly from the gradient domination property in the single MDP) implies that any stationary point of $V_s^i$ (w.r.t the variables $x_{i,s,a}$ of agent's $i$ policy with the rest of the variables being fixed) is an optimal policy for $i$, i.e., a best response given the policies of all other agents.

Lemma D.3 also suggests that there is an important difference in the gradient domination property between (multi-agent) MPGs and single agent MDPs. Specifically, for MPGs, the value of each agent at different Nash policies may not be unique[6] which implies that the gradient domination property, as stated in Lemma D.3, will only be enough to guarantee convergence to *one of* the optimal (stationary) points of $\Phi$ (and not necessarily to the absolute maximum of $\Phi$).

The last critical step before the formal statement and proof of Theorem 1.2 is to show that the potential function $\Phi$ is smooth. Importantly, the smoothness parameter, involves $A_{\max} := \max_{i \in \mathcal{N}} |\mathcal{A}_i|$, i.e., the maximum number of actions for some agent, which scales *linearly* in the number of agents.

**Lemma D.4** (Smoothness of $\Phi$). *Let $A_{\max} := \max_{i \in \mathcal{N}} |\mathcal{A}_i|$ (the maximum number of actions for some agent). Then, for any initial state $s_0 \in \mathcal{S}$ (and hence for every distribution $\mu \in \Delta(\mathcal{S})$ on states), $\Phi_\mu(\pi)$ is $\frac{2n\gamma A_{\max}}{(1-\gamma)^3}$-smooth, i.e., it holds that*

$$\left\| \nabla_\pi \Phi_{s_0}(\pi) - \nabla_\pi \Phi_{s_0}(\pi') \right\|_2 \leq \frac{2n\gamma A_{\max}}{(1 - \gamma)^3} \left\| \pi - \pi' \right\|_2 \tag{13}$$

*Proof of Lemma D.4.* It suffices to show that the maximum eigenvalue in absolute value of the Hessian of $\Phi$ is at most $\frac{2n\gamma A_{\max}}{(1-\gamma)^3}$, i.e., that

$$\left\| \nabla^2 \Phi_\mu \right\|_2 \leq \frac{2n\gamma A_{\max}}{(1 - \gamma)^3} .$$

---

[6]This is in contrast to single-agent MDPs for which the agent has a unique optimal value even though their optimal policy may not necessarily be unique.

We first prove the following intermediate claim.

**Claim D.5.** *Consider the symmetric block matrix $C$ with $n \times n$ matrices so that $\|C_{ij}\|_2 \leq L$. Then, it holds that $\|C\|_2 \leq nL$, i.e., if all block submatrices have spectral norm at most $L$, then $C$ has spectral norm at most $nL$.*

*Proof.* We will prove the claim by induction on $n$. For $n = 2$ we need to show that

$$\|C\|_2 := \left\| \begin{pmatrix} C_{11} & C_{12} \\ C_{21} & C_{22} \end{pmatrix} \right\|_2 \leq 2L$$

if $\|C_{11}\|_2, \|C_{12}\|_2, \|C_{21}\|_2, \|C_{22}\|_2 \leq L$. Define matrix $W$ to be

$$W := 2L \cdot I - C = \begin{pmatrix} 2L \cdot I - C_{11} & -C_{12} \\ -C_{21} & 2L \cdot I - C_{22} \end{pmatrix},$$

where $I$ is the identity matrix (of appropriate size). If we show that $W$ is positive semi-definite, then it follows that $W$ has only non-negative eigenvalues, which, in turn, implies that the spectral norm of $C$ is at most $2L$. To see this, set

$$W_1 := \begin{pmatrix} L \cdot I - C_{11} & 0 \\ 0 & L \cdot I - C_{22} \end{pmatrix}, W_2 := \begin{pmatrix} L \cdot I & -C_{12} \\ -C_{21} & L \cdot I \end{pmatrix}.$$

$W_1$ is positive semi-definite as a block diagonal matrix with diagonal blocks positive semi-definite matrices. Moreover, by Schur complement we get that $W_2$ is positive semi-definite as long as $L \cdot I$ is positive semi-definite and $L \cdot I - \frac{1}{L} \cdot C_{12}C_{21}$ is positive semidefinite. By assumption, we have that

$$\frac{1}{L} \|C_{12}C_{21}\|_2 \leq \frac{1}{L} \|C_{12}\|_2 \|C_{12}\|_2 \leq L,$$

which implies that $L \cdot I - \frac{1}{L} \cdot C_{12}C_{21}$ has non-negative eigenvalues. Thus, $W_2$ is positive semi-definite. We conclude that $W_1 + W_2$ is positive semi-definite (sum of positive semi-definite matrices is positive semi-definite) and the claim follows.

For the induction step, suppose that the claim holds for an $n = k - 1 \geq 2$. To establish that it also holds for $k$, we need to show that

$$\|C\|_2 := \left\| \begin{pmatrix} C_{11} & C_{12} & \ldots & C_{1k} \\ C_{21} & C_{22} & \ldots & C_{2k} \\ \vdots & \vdots & \vdots & \vdots \\ C_{k1} & C_{k2} & \ldots & C_{kk} \end{pmatrix} \right\|_2 \leq kL$$

as long as $\|C_{ij}\|_2 \leq L$ for all $i, j$. Let $W = kL \cdot I - C$. To show that $W$ is positive semi-definite consider

$$W_1 := \begin{pmatrix} kL \cdot I - C_{11} & -C_{12} & -C_{13} & \ldots & -C_{1k} \\ -C_{21} & L \cdot I & 0 & \ldots & 0 \\ \vdots & \vdots & \vdots & \vdots & \\ -C_{k1} & 0 & 0 & \ldots & L \cdot I \end{pmatrix}, W_2 := W - W_1.$$

By induction, it follows that $W_2$ is positive semi-definite. We need to show that the same holds for $W_1$. By Schur complement we obtain that $W_1$ is positive semi-definite if and only if $kL \cdot I - C_{11} - \frac{1}{L} \sum_{i=2}^{k} C_{1i}C_{i1}$ is positive semi-definite. It follows that

$$\left\| C_{11} - \frac{1}{L} \sum_{i} C_{1i}C_{i1} \right\|_2 \leq \|C_{11}\|_2 + \frac{1}{L} \sum_{i=2}^{k} \|C_{1i}\|_2 \|C_{i1}\|_2 \leq L + (k-1)L = kL.$$

Hence $W_1$ is positive semi-definite and the induction is complete. $\qquad\square$

Returning to the statement of Lemma D.4, we will show that

$$\left\| \nabla^2_{\pi_j \pi_i} V^j_\mu \right\|_2 \leq C, \tag{14}$$

for all $i, j \in \mathcal{N}$ with $C$ chosen to be $\frac{2\gamma A_{\max}}{(1-\gamma)^3}$. Assuming we have shown (14), we conclude from Claim D.5 that $\left\| \nabla^2 \Phi_\mu \right\|_2 \leq nC$, and hence $\Phi$ will be $nC$-smooth (the proof of Lemma D.4 will follow).

To prove (14), we follow the same proof steps as in the proof of Agarwal et al. (2020), Lemma D.3. We will need to prove an upper bound on the largest eigenvalue (in absolute value) of the matrix

$$\nabla^2_{\pi_j \pi_i} V^j_\mu = \nabla^2_{\pi_j \pi_i} V^i_\mu,$$

along the direction where only agent $i$ is allowed to change policy.

Fix policy $\pi = (\pi_1, ..., \pi_n)$, agents $i \neq j$, scalars $t, s \geq 0$, state $s_0$ and $u, v$ be unit vectors such that $\pi_i + t \cdot u \in \Delta(\mathcal{A}_i)^S$ and $\pi_j + s \cdot v \in \Delta(\mathcal{A}_j)^S$. Moreover, let $V(t) = V^i_{s_0}(\pi_i + t \cdot u, \pi_{-i})$. and $W(t, s) = V^i_{s_0}(\pi_i + t \cdot u, \pi_j + s \cdot v, \pi_{-i,-j})$. It suffices to show that

$$\max_{\|u\|_2=1} \left| \frac{d^2 V(0)}{dt^2} \right| \leq \frac{2\gamma|\mathcal{A}_i|}{(1-\gamma)^3} \quad \text{and} \quad \max_{\|u\|_2=1} \left| \frac{d^2 W(0,0)}{dtds} \right| \leq \frac{2\gamma\sqrt{|\mathcal{A}_i||\mathcal{A}_j|}}{(1-\gamma)^3}. \tag{15}$$

- We first focus on $V(t)$. It holds that

$$V(t) = \sum_{a \in \mathcal{A}_i} \sum_{\mathbf{a} \in \mathcal{A}_{-i}} (x_{i,s_0,a} + tu_{i,s_0,a}) \prod_{j \neq i} x_{j,s_0,a_j} Q^i_{s_0}((\pi_i + tu, \pi_{-i}), (a, \mathbf{a}))$$

(note that $\sum_{a \in \mathcal{A}_i} \sum_{\mathbf{a} \in \mathcal{A}_{-i}} (x_{i,s_0,a} + tu_{i,s_0,a}) \prod_{j \neq i} x_{j,s_0,a_j} = 1$ since it is a distribution), hence taking the second derivative we have

$$\begin{aligned}
\frac{d^2 V(0)}{dt^2} &= \sum_{a \in \mathcal{A}_i} \sum_{\mathbf{a} \in \mathcal{A}_{-i}} (x_{i,s_0,a} + tu_{i,s_0,a}) \prod_{j \neq i} x_{j,s_0,a_j} \frac{d^2 Q^i_{s_0}(\pi, (a, \mathbf{a}))}{dt^2} \\
&\quad + 2 \sum_{a \in \mathcal{A}_i} \sum_{\mathbf{a} \in \mathcal{A}_{-i}} u_{i,s_0,a} \prod_{j \neq i} x_{j,s_0,a_j} \frac{dQ^i_{s_0}(\pi, (a, \mathbf{a}))}{dt}
\end{aligned} \tag{16}$$

For the remaining of the first part of the proof, we shall show

$$\left| \frac{dQ^i_{s_0}(\pi, (a, \mathbf{a}))}{dt} \right| \leq \frac{\gamma\sqrt{|\mathcal{A}_i|}}{(1-\gamma)^2} \quad \text{and} \quad \left| \frac{d^2 Q^i_{s_0}(\pi, (a, \mathbf{a}))}{dt^2} \right| \leq \frac{2\gamma^2|\mathcal{A}_i|}{(1-\gamma)^3},$$

and then combining with (16) we get

$$\left| \frac{d^2 V(0)}{dt^2} \right| \leq \frac{2\gamma\sqrt{|\mathcal{A}_i|}}{(1-\gamma)^2} \sum_{a \in \mathcal{A}_i} |u_{i,s_0,a}| + \frac{2\gamma^2|\mathcal{A}_i|}{(1-\gamma)^3} \leq \frac{2\gamma|\mathcal{A}_i|}{(1-\gamma)^2} + \frac{2\gamma^2|\mathcal{A}_i|}{(1-\gamma)^3} \leq \frac{2\gamma A_{\max}}{(1-\gamma)^3}.$$

To bound the derivative of the $Q$-function, observe that $Q^i_{s_0}((\pi_i + tu, \pi_{-i}), (a, \mathbf{a})) = e^\top_{s_0,a}(I - \gamma P(t))^{-1} r$, where $r(s_0, a)$ is the expected reward of agent $i$ (w.r.t the randomness of the remaining agents) if he chooses action $a$ at state $s_0$ and $P(t)$ is state-action transition matrix of w.r.t the joint distribution of all agents but $i$, i.e., $\pi_{-i}$ and the environment.

It is clear that $\frac{d^2 P}{dt^2} = 0$ (linear with respect to $t$ due to the direct parameterization) and that $\left\| \frac{dP}{dt} \right\|_\infty \leq \sum_{a \in \mathcal{A}_i} |u_{i,s_0,a}| \leq \sqrt{\mathcal{A}_i} \leq \sqrt{A_{\max}}$. Using that $\left\| (I - \gamma P(t))^{-1} \right\|_\infty \leq \frac{1}{1-\gamma}$, we get

$$\left| \frac{dQ^i_{s_0}(\pi, (a, \mathbf{a}))}{dt} \right| = \gamma \left| e^\top_{s_0,a}(I - \gamma P(0))^{-1} \frac{dP(0)}{dt}(I - \gamma P(0))^{-1} r \right| \leq \frac{\gamma\sqrt{|\mathcal{A}_i|}}{(1-\gamma)^2} \leq \frac{\gamma\sqrt{A_{\max}}}{(1-\gamma)^2}, \tag{17}$$

and also

$$\begin{aligned}
\left| \frac{d^2 Q^i_{s_0}(\pi, (a, \mathbf{a}))}{dt^2} \right| &= 2\gamma^2 \left| e^\top_{s_0,a}(I - \gamma P(0))^{-1} \frac{dP(0)}{dt}(I - \gamma P(0))^{-1} \frac{dP(0)}{dt}(I - \gamma P(0))^{-1} r \right| \\
&\leq \frac{2\gamma^2|\mathcal{A}_i|}{(1-\gamma)^3} \leq \frac{2\gamma^2 A_{\max}}{(1-\gamma)^3},
\end{aligned} \tag{18}$$

Since $u$ is arbitrary, the first part of (15) is proved.

- For the second part, we focus on $W(t)$ which is equal to

$$
\begin{aligned}
W(t,s) = \sum_{a\in\mathcal{A}_i}\sum_{b\in\mathcal{A}_j}\sum_{\mathbf{a}\in\mathcal{A}_{-i,-j}} & (x_{i,s_0,a}+tu_{i,s_0,a})(x_{j,s_0,b}+sv_{j,s_0,b})\cdot \\
& \cdot \prod_{j'\neq i,j} x_{j',s_0,a_{j'}} Q_{s_0}^i((\pi_i+tu,\pi_j+sv,\pi_{-i,-j}),(a,b,\mathbf{a}))
\end{aligned}
\tag{19}
$$

We consider the derivative of $W$ (19) and we get

$$
\begin{aligned}
\frac{dW(0,0)}{dtds} = & \sum_{a\in\mathcal{A}_i}\sum_{b\in\mathcal{A}_j}\sum_{\mathbf{a}\in\mathcal{A}_{-i,-j}} u_{i,s_0,a}v_{j,s_0,b}\cdot\prod_{j'\neq i,j}x_{j',s_0,a_{j'}}Q_{s_0}^i(\pi,(a,b,\mathbf{a})) \\
& + \sum_{a\in\mathcal{A}_i}\sum_{\mathbf{a}\in\mathcal{A}_{-i}}u_{i,s_0,a}\cdot\prod_{j'\neq i}x_{j',s_0,a_{j'}}\frac{dQ_{s_0}^i(\pi,(a,\mathbf{a}))}{dt} \\
& + \sum_{b\in\mathcal{A}_j}\sum_{\mathbf{a}\in\mathcal{A}_{-j}}v_{j,s_0,b}\cdot\prod_{j'\neq j}x_{j',s_0,a_{j'}}\frac{dQ_{s_0}^i(\pi,(b,\mathbf{a}))}{dt} + \sum_{\mathbf{a}\in\mathcal{A}}\prod_{j'}x_{j',s_0,a_{j'}}\frac{d^2Q_{s_0}^i(\pi,\mathbf{a})}{dtds}.
\end{aligned}
$$

The first term of the sum in absolute value is at most $\frac{\sqrt{|\mathcal{A}_i||\mathcal{A}_j|}}{1-\gamma}$ (assuming rewards lie in $[0,1]$.)
Moreover, using (17) the second term of the sum in absolute value is bounded by $\frac{\gamma\sqrt{|\mathcal{A}_i|}\sqrt{|\mathcal{A}_i|}}{(1-\gamma)^2}$ and
the third term by $\frac{\gamma\sqrt{|\mathcal{A}_j|}\sqrt{|\mathcal{A}_i|}}{(1-\gamma)^2}$. To bound the $\frac{d^2Q_{s_0}^i(\pi,\mathbf{a})}{dtds}$, the same approach works that we used
to prove (18) with the extra fact that the state-action transition matrix is $P(t,s)$ and moreover, the
reward $r(s_0,a,b)$ is the expected reward of agent $i$ (w.r.t the randomness of all agents but $i,j$) if $i$
chooses action $a$ and $j$ chooses $b$ at state $s_0$. Finally, for the fourth term we get that

$$
\left|\frac{d^2Q_{s_0}^i(\pi,\mathbf{a})}{dtds}\right|\leq
$$

$$
\leq \gamma^2\left|e_{s_0,a}^\top(I-\gamma P(0,0))^{-1}\frac{dP(0,0)}{ds}(I-\gamma P(0,0))^{-1}\frac{dP(0,0)}{dt}(I-\gamma P(0,0))^{-1}r\right| +
$$

$$
+ \gamma^2\left|e_{s_0,a}^\top(I-\gamma P(0,0))^{-1}\frac{dP(0,0)}{dt}(I-\gamma P(0,0))^{-1}\frac{dP(0,0)}{ds}(I-\gamma P(0,0))^{-1}r\right| +
$$

$$
+ \gamma\left|e_{s_0,a}^\top(I-\gamma P(0,0))^{-1}\frac{d^2P(0,0)}{dt^2}(I-\gamma P(0,0))^{-1}r\right|
$$

$$
\leq \frac{\gamma^2\sqrt{|\mathcal{A}_i||\mathcal{A}_j|}}{(1-\gamma)^3} + \frac{\gamma^2\sqrt{|\mathcal{A}_i||\mathcal{A}_j|}}{(1-\gamma)^3} + \frac{\gamma\sqrt{|\mathcal{A}_i||\mathcal{A}_j|}}{(1-\gamma)^2} \leq \frac{2\gamma\sqrt{|\mathcal{A}_i||\mathcal{A}_j|}}{(1-\gamma)^3} \leq \frac{2\gamma A_{\max}}{(1-\gamma)^3}. \qquad \Box
$$

*Proof of Theorem 4.2.* The first step is to show that $\Phi$ is a $\beta$-smooth function, in particular, that $\nabla_\pi\Phi$
is $\beta$-Lipschitz with $\beta=\frac{2n\gamma A_{\max}}{(1-\gamma)^3}$ as established in Lemma D.4. Then, a standard *ascent lemma* for
Gradient Ascent (see Lemma C.1 from Bubeck (2015)) implies that for any $\beta$-smooth function $f$ it
holds that $f(x')-f(x)\geq\frac{1}{2\beta}\|x'-x\|_2^2$ where $x'$ is the next iterate of equation PGA. Applied to
our setting, this gives

$$
\Phi_\mu(\pi^{(t+1)})-\Phi_\mu(\pi^{(t)})\geq\frac{(1-\gamma)^3}{4n\gamma A_{\max}}\left\|\pi^{(t+1)}-\pi^{(t)}\right\|_2^2
\tag{20}
$$

Thus, if the number of iterates, $T$, is $\frac{16n\gamma D^2SA_{\max}}{(1-\gamma)^5\epsilon^2}$, then there must exist a $1\leq t\leq T$ so that
$\left\|\pi^{(t+1)}-\pi^{(t)}\right\|_2\leq\frac{\epsilon(1-\gamma)}{2D\sqrt{S}}$. Using a standard approximation property (see Lemma C.2), we then
conclude that $\pi^{(t+1)}$ will be a $\frac{\epsilon(1-\gamma)}{D\sqrt{S}}$-stationary point for the potential function $\Phi$. Hence, by
Lemma 4.1, it follows that $\pi^{(t+1)}$ is an $\epsilon$-Nash policy and the proof is complete. $\qquad\Box$

## D.2 FINITE SAMPLE CASE

*Proof of Lemma 4.3.* It is straightforward from Lemma C.4 and the equality of the partial derivatives
between the value functions and the potential, i.e., $\nabla_{\pi_i}\Phi_\mu=\nabla_{\pi_i}V_\mu^i$ for all $i\in\mathcal{N}$ (see property P2
in Proposition B.1). $\qquad\Box$

*Proof of Theorem 4.4.* Let $\delta_t = \hat{\nabla}_\pi^{(t)} - \nabla_\pi \Phi_\mu(\pi^{(t)})$ and set $\lambda = \frac{(1-\gamma)^3}{2n\gamma A_{\max}}$ (the inverse of the smooth parameter in D.4). Moreover, we set $y^{(t+1)} = P_{\Delta(\mathcal{A})^S}(\pi^{(t)} + \eta\nabla_\pi\Phi(\pi^{(t)}))$ ($y^{(t+1)}$ captures the next iterate of the projected (deterministic) gradient ascent). We follow the analysis of Projected Stochastic Gradient Ascent for non-convex smooth-functions (see Davis & Drusvyatskiy (2018), Theorem 2.1) that makes use of the Moreau envelope. Let

$$\phi_\lambda(x) = \underset{y \in \Delta(\mathcal{A})^S}{\arg\min}\left\{-\Phi_\mu(y) + \frac{1}{\lambda}\|x - y\|_2^2\right\},$$

(definition of Moreau envelope for our objective $\Phi$). From the definition of $\phi$ and a standard property of projection we get

$$
\begin{aligned}
\phi_\lambda(\pi^{(t+1)}) &\leq -\Phi_\mu(y^{(t+1)}) + \frac{1}{\lambda}\left\|\pi^{(t+1)} - y^{(t+1)}\right\|_2^2 \\
&\leq -\Phi_\mu(y^{(t+1)}) + \frac{1}{\lambda}\left\|\pi^{(t)} + \eta\hat{\nabla}_\pi^{(t)} - y^{(t+1)}\right\|_2^2 \qquad (21) \\
&= -\Phi_\mu(y^{(t+1)}) + \frac{1}{\lambda}\left\|\pi^{(t)} - y^{(t+1)}\right\|_2^2 + \frac{\eta^2}{\lambda}\left\|\hat{\nabla}_\pi^{(t)}\right\|_2^2 + \frac{2\eta}{\lambda}(\pi^{(t)} - y^{(t+1)})^\top \hat{\nabla}_\pi^{(t)}
\end{aligned}
$$

Since $\hat{\nabla}_\pi^{(t)}$ is unbiased (Lemma 4.3) we have that $\mathbb{E}[\delta_t|\pi^{(t)}] = 0$, therefore $\mathbb{E}\left[\delta_t^\top(y^{(t+1)} - \pi^{(t)})\right] = 0$. Additionally, by Lemma 4.3 (applied for all agents $i$) we also have $\mathbb{E}\left[\left\|\hat{\nabla}_\pi^{(t)}\right\|_2^2\right] \leq \frac{24nA_{\max}^2}{\epsilon(1-\gamma)^4}$. Hence by taking expectation on (21) we have:

$$
\begin{aligned}
\mathbb{E}[\phi_\lambda(\pi^{(t+1)})] &\leq \\
&\leq \mathbb{E}\left[-\Phi_\mu(y^{(t+1)}) + \frac{1}{\lambda}\left\|\pi^{(t)} - y^{(t+1)}\right\|_2^2\right] + \frac{2\eta}{\lambda}\mathbb{E}[(\pi^{(t)} - y^{(t+1)})^\top\nabla_\pi\Phi_\mu(\pi^{(t)})] + \frac{24\eta^2 nA_{\max}^2}{\lambda\epsilon(1-\gamma)^4}.
\end{aligned}
$$

Using the definition of Moreau envelope and the fact that $\Phi$ is $\frac{1}{\lambda}$-smooth (Lemma D.4, after the parameterization, the smoothness parameter does not increase) we conclude that

$$
\begin{aligned}
\mathbb{E}[\phi_\lambda(\pi^{(t+1)})] &\leq \\
&\leq \mathbb{E}[\phi_\lambda(\pi^{(t)})] + \frac{2\eta}{\lambda}\mathbb{E}[(\pi^{(t)} - y^{(t+1)})^\top\nabla_\pi\Phi_\mu(\pi^{(t)})] + \frac{24\eta^2 nA_{\max}^2}{\lambda\epsilon(1-\gamma)^4} \\
&\leq \mathbb{E}[\phi_\lambda(\pi^{(t)})] + \frac{2\eta}{\lambda}\mathbb{E}\left[\Phi_\mu(\pi^{(t)}) - \Phi_\mu(y^{(t+1)}) + \frac{1}{2\lambda}\left\|\pi^{(t)} - y^{(t+1)}\right\|_2^2\right] + \frac{24\eta^2 nA_{\max}^2}{\lambda\epsilon(1-\gamma)^4},
\end{aligned}
$$

or equivalently

$$
\begin{aligned}
\mathbb{E}[\phi_\lambda(\pi^{(t+1)})] &- \mathbb{E}[\phi_\lambda(\pi^{(t)})] \leq \\
&\leq \frac{2\eta}{\lambda}\mathbb{E}\left[\Phi_\mu(\pi^{(t)}) - \Phi_\mu(y^{(t+1)}) + \frac{1}{2\lambda}\left\|\pi^{(t)} - y^{(t+1)}\right\|_2^2\right] + \frac{24\eta^2 nA_{\max}^2}{\lambda\epsilon(1-\gamma)^4} \qquad (22)
\end{aligned}
$$

Adding telescopically (22), dividing by $T$ and because w.l.o.g $-\Phi \in [-1, 0]$, we get that

$$
\begin{aligned}
\frac{1}{T} + \frac{24\eta^2 nA_{\max}^2}{\lambda\epsilon(1-\gamma)^4} &\geq \frac{2\eta}{\lambda T}\sum_{t=1}^T \mathbb{E}\left[\Phi_\mu(y^{(t+1)}) - \Phi_\mu(\pi^{(t)})\right] - \frac{\eta}{\lambda^2 T}\sum_{t=1}^T\mathbb{E}\left[\left\|y^{(t+1)} - \pi^{(t)}\right\|_2^2\right] \\
&\geq \min_{t\in[T]}\left\{\frac{2\eta}{\lambda}\mathbb{E}\left[\Phi_\mu(y^{(t+1)}) - \Phi_\mu(\pi^{(t)})\right] - \frac{\eta}{\lambda^2}\mathbb{E}[\left\|y^{(t+1)} - \pi^{(t)}\right\|_2^2]\right\} \qquad (23)
\end{aligned}
$$

Let $t*$ be the time index that minimizes the above. We show the following inequality (which provides a lower bound on the RHS of (23)):

$$\mathbb{E}\left[\Phi_\mu(y^{(t*+1)}) - \Phi_\mu(\pi^{(t*)})\right] - \frac{1}{2\lambda}\mathbb{E}\left[\left\|y^{(t*+1)} - \pi^{(t*)}\right\|_2^2\right] \geq \frac{1}{\lambda}\mathbb{E}\left[\left\|y^{(t*+1)} - \pi^{(t*)}\right\|_2^2\right] \qquad (24)$$

Observe that by $\frac{1}{\lambda}$-smoothness of $\Phi_\mu$ we get that $H(x) := -\Phi_\mu(x) + \frac{1}{\lambda}\left\|x - \pi^{(t*)}\right\|_2^2$ is $\frac{1}{\lambda}$-strong convex and moreover, $y^{(t+1)}$ is the minimizer of $H$. Therefore, we get that $H(\pi^{t*}) - H(y^{t*+1}) \geq$

$\frac{1}{2\lambda}\left\|\pi^{t*}-y^{t*+1}\right\|_2^2$, or equivalently

$$\Phi_\mu(y^{t*+1}) - \Phi_\mu(\pi^{t*}) - \frac{1}{\lambda}\left\|\pi^{t*}-y^{t*+1}\right\|_2^2 \geq \frac{1}{2\lambda}\left\|\pi^{t*}-y^{t*+1}\right\|_2^2.$$

By taking expectation of terms above, (24) follows. Combining (23) with (24) we conclude that

$$\frac{1}{T} + \frac{24\eta^2 n A_{\max}^2}{\lambda\epsilon(1-\gamma)^4} \geq \frac{2\eta}{\lambda^2}\mathbb{E}\left[\left\|\pi^{t*}-y^{t*+1}\right\|_2^2\right].$$

By Jensen's inequality it occurs that

$$\mathbb{E}\left[\left\|y^{(t*+1)}-\pi^{(t*)}\right\|_2\right] \leq \sqrt{\frac{\lambda^2}{2\eta T} + \eta\frac{12n\lambda A_{\max}^2}{\epsilon(1-\gamma)^4}}. \tag{25}$$

To get an $\epsilon$-Nash policy, we have to bound $\left\|y^{(t*+1)}-\pi^{(t*)}\right\|_2 \leq \frac{\epsilon(1-\gamma)}{2D\sqrt{S}(2+\sqrt{nA_{max}})}$ and choose $\alpha = \epsilon^2$ in the greedy parameterization. This is true because of Lemma C.3 Lemma 4.1. Hence, we need to choose $\eta, T$ so that

$$\sqrt{\frac{\lambda^2}{2\eta T} + \eta\frac{12n\lambda A_{\max}^2}{\epsilon^2(1-\gamma)^4}} \leq \frac{\epsilon(1-\gamma)}{2D\sqrt{S}(2+\sqrt{nA_{max}})}.$$

We conclude that $\eta$ can be chosen to be $\frac{\epsilon^4(1-\gamma)^3\gamma}{48nD^2A_{\max}^2 S}$ and $T$ to be $\frac{48(1-\gamma)A_{\max}D^4 S^2}{\epsilon^6\gamma^3}$. $\qquad\square$

## E  ADDITIONAL EXPERIMENTS

In this part, we analyze variations of the experimental setting in Section 5.

**Coordination beyond MPGs.**  As mentioned in Remark 1, we know that policy gradient converges also in other cooperative settings (e.g., in weighted and asymptotically also in ordinal MPGs). To study such cases, we modify our experiment from Section 5. The setting remains mostly the same, except now in the distancing state the rewards are reduced by a differing (yet still sufficiently large) amount, $c_k$, for each facility $k = A, B, C, D$. Despite this asymmetry, agents still have aligned incentives. In fact, if the $c_k$'s for all $k = A, B, C, D$ are taken to be greater than $c$ in the MDP from Section 5, then the agents can be said to have even "stronger" incentive to cooperate. The results of running independent policy gradient on this variant are shown in Figure 6. Despite requiring more iterations compared to the symmetric setting, the algorithm still converges to the same Nash policy.

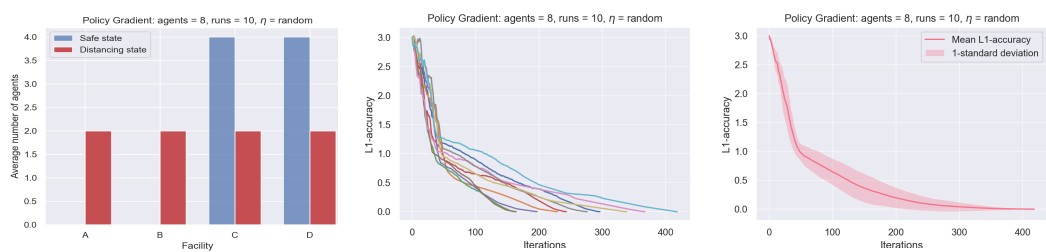

Figure 6: Figures similar to Figure 5, except now with $c > c_A > c_B > c_C > c_D$ as described in the text. Independent policy gradient requires more iterations to converge compared to the symmetric shift setting, but still arrives at the same Nash policy.

**Coordination with more agents and facilities.**  We next test the performance of the independent policy gradient algorithm in a larger setting with $N = 16$ agents and $A_i = 5$ facilities, $\mathcal{A}_i = \{A, B, C, D, E\}$ with $w_A < w_B < w_C < w_D < w_E$ (i.e., $E$ is the most preferable by all agents). We use a learning rate $\eta = 0.0001$ for all agents (which is again much larger than the theoretical guarantee of Theorem 4.2). All runs lead to convergence to an (optimal) Nash policy as shown in the middle and rightmost panels. The leftmost panel shows the distribution of the agents among facilities in both states, which is the same (and the optimal one) in all Nash policies that are reached by the algorithm. The results are shown in Figure 7.

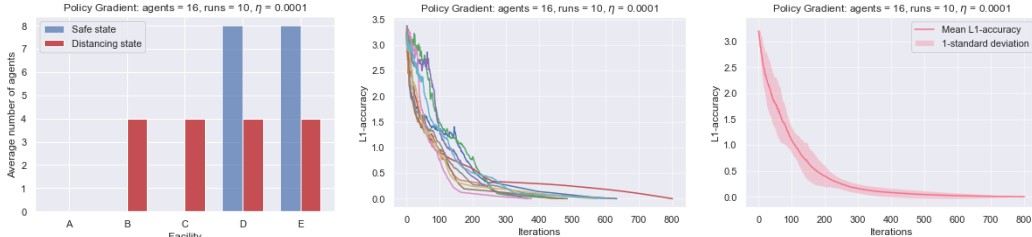

Figure 7: Convergence to deterministic Nash policies of independent policy gradient in a variation of the MDP of Section 5 with $N = 16$ agents and $A_i = 5$ facilities, $\mathcal{A}_i = \{A, B, C, D, E\}$ with $w_A < w_B < w_C < w_D < w_E$ (i.e., $E$ is the most preferable by all agents). Again, while there are several (symmetric) deterministic Nash policies, all of them yield the same distribution of agents among states (leftmost panel). All runs converge successfully to that outcome.

**Coordination with random transitions**   Next, we study the effect of adding randomness to the transitions on the performance of the individual policy gradient algorithm. In this case, we experiment with the same setting as in Section 5 (i.e., $N = 8$ agents and $A_i = 4$ facilities that each agent $i \in \mathcal{N}$ can choose from), but use the following stochastic transition rule instead: in addition to the existing transition rules, the sequence of play may transition from the safe to the distancing state with probability $p\%$ regardless of the distribution of the agents and may remain at the distancing state with probability $q\%$ again regardless of the distribution of the agents there.

Two sets of results are presented in Figure 8. In the first (upper panels), we use $p, q = 1\%, 10\%$ and in the second $p, q = 5\%, 20\%$. In both cases, we use a learning rate $\eta = 0.0001$ (several orders of magnitude higher than what is required by Theorem 4.4). Independent policy gradient converges in both cases to deterministic Nash policies despite the randomness in the transitions. However, for higher levels of randomness (lower panels), the algorithm remains at an $\epsilon$-Nash policy for a high number of iterations. This is in line with the theoretical predictions of Theorem 4.4.

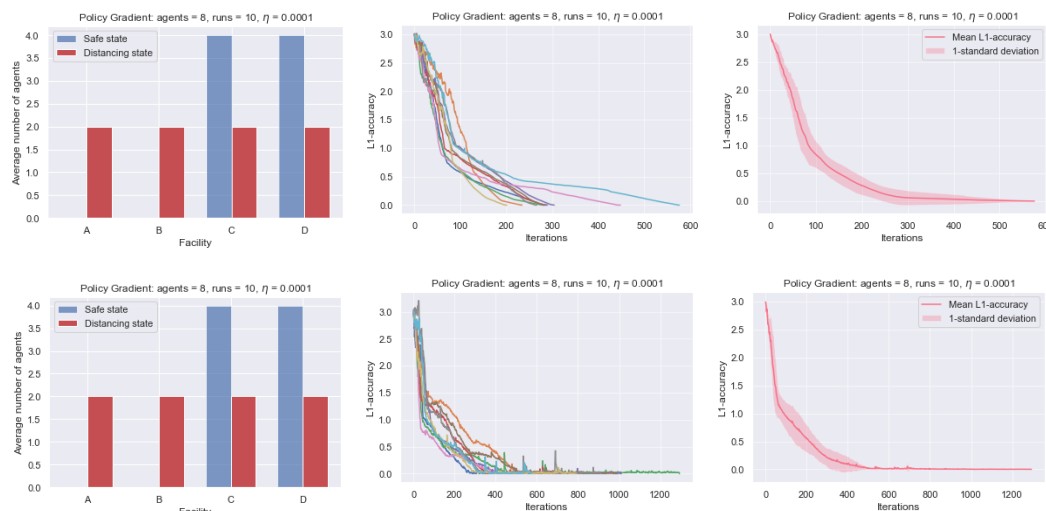

Figure 8: Convergence to deterministic Nash policies of independent policy gradient in two variations of the MDP of Section 5 with stochastic transitions between states.

**Independent Policy Gradient in a Different Environment.**   A subsequent work to this paper introduced a different multi-agent MDP environment that they prove is an MPG, which they refer to as *stochastic congestion games* (Fox et al., 2021). This stochastic congestion or routing game was already well-studied before this paper, especially in Mguni et al. (2021), so it is valuable to see that this MPG framework captures such games. In this environment, there is an underlying directed acyclic graph $G$ that has a source node $s$ and a sink node $t$. There are $N$ agents. The actual states

of the finite-horizon MDP are all the possible positions of the agents on the graph. When the state consisting of all agents being at $t$ is reached, the MDP terminates. At each time step, each agent chooses an action, which in this case is an edge $e$ to transverse, and the cost they experience is a function of the load of the edge, which is the total number of agents choosing edge $e$. A more complete description and proof that this environment is an MPG is provided in Fox et al. (2021).

We implemented this environment with $N = 4$ agents, matching the number of agents in Fox et al. (2021), but we were actually able to increase the size of the underlying graph from the 6 vertices done in that paper to 8. The game is visualized in Figure 9.

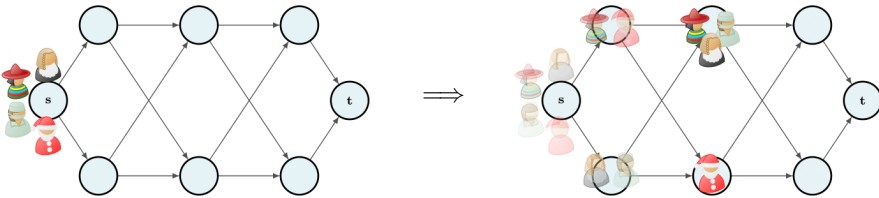

Figure 9: The stochastic congestion game with $N = 4$ agents and 8 nodes. In this discrete time MPG, all agents start at $s$ and at each point in time, they move to one of the nodes of the next layer. Their goal is to reach $t$ at a minimum cost (as expressed by the congestion on the the vertices). The left panel shows the starting state and the right panel shows one possible state after two transitions (with intermediate states in pale colors).

We used our implementation of the independent policy gradient algorithm with the same parameters as in our experiment from Section 5, specifically we have $T = 20$, $\gamma = 0.99$, and $\eta = 0.0001$. The results are shown in Figure 10.

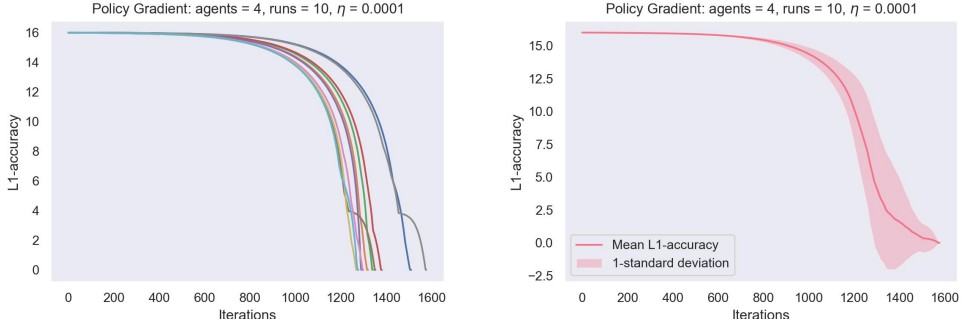

Figure 10: Convergence of independent policy gradient in the stochastic congestion game with $N = 4$ and 6 vertices between source and target (3 layers of 2 vertices each) as shown in Figure 9.

