# OpenReview forum: "Global Convergence of Multi-Agent Policy Gradient in Markov Potential Games"
_ICLR.cc/2022/Conference — ICLR 2022 Poster_

### Official Review · Reviewer_idii · 2021-10-31

**Correctness:** 4
**Technical Novelty And Significance:** 3
**Empirical Novelty And Significance:** Not applicable
**Recommendation:** 8
**Confidence:** 3

**Main Review:**

Strength: This paper considers an important research topic --- policy gradient in markov potential games (MPG) and provides interesting new insights on this topic. The existence of deterministic Nash policies in MPG is somewhat not surprising though, I find the examples 1 and 2, which shows the potentiality at every state may not lead to MPG, illuminating. The methodology for analyzing the convergence property of the projected (stochastic) policy gradient seems to be adapted from some previous works. However, in general, the presented convergence results in MPG are novel.

Weakness: Some related works have not been discussed in the paper. For example, the paper "Stochastic Potential Games" (https://arxiv.org/pdf/2005.13527v1.pdf) also considers potential games with multiple states. I wonder how Theorem 3.1 of this paper is different from Theorem 1 of "Stochastic Potential Games." The paper is generally well-written, but some parts remain unclear:
1. For Condition C2 of Proposition 3.2,  the authors state that "the dummy terms of each agent's immediate rewards are equal across all states." Intuitively, this seems to suggest that $u_s^i(\vec{a}_{-i})$ equals some constant for all states, which is not in line with the condition formalized right after the statement. So how to interpret the formalism of C2? I think a more clear and precise interpretation is needed.
2. The example provided in Figure 3 assumes that $p_0$ does not depend on agents' actions. Suppose there is a 2-player MDP which is not potential at every state and $p_0$ depends on agents' actions, is it possible that this 2-player MDP is an MPG?
3. Section 4 is rather dense and not easy to follow. In the meantime, Section 5 does not provide much new knowledge about congestion games. The authors may consider moving some parts of Section 5 to the Appendix and providing more explanations (such as the definitions and prior results required to derive Theorem 4.2) in Section 4.


**Summary Of The Paper:**

This paper examines the projected policy gradient method in markov potential games (MPG). The paper i) proves the existence of deterministic Nash policies in MPG, ii) analyzes the sufficient conditions for MPG, and iii) proves the convergece of projected (stochastic) gradient ascent to Nash policies in MPG.

**Summary Of The Review:**

This paper provides interesting, new, and important theoretical results for policy gradient in markov potential games.

---

> ### Author Response · Authors · 2021-11-19
> **Response to Reviewer idii**
>
> We thank reviewer idii for their support and constructive feedback on our paper.
> - We now discuss in the Other Related Works section the connection between our Theorem 3.1 and Theorem 3.1 in version 1 of the "Stochastic Potential Games" (SPG) paper. In particular, we highlight that the results are analogous but in different settings, since the SPG paper assumes the state-transitivity condition.
> 1. We thank the reviewer for pointing out the lack of clarity in condition C2 of Proposition 3.2. We have rephrased accordingly. As an additional explanation (not provided in the paper), we note that the condition of constant (rather than zero) gradients stems directly from the KKT optimization conditions in the non-binding interior of the simplex.
> 2. The only reason that we assume $p_0$ to be action-independent is to maintain the attention at the construction within the dotted rectangle by making the rest of the example as simple as possible. In other words, there is nothing special about the selection of $p_0$ to be action independent. In fact, it shows that an MDP with a zero-sum state can be MPG even in the simple possible setting (however, the transitions inside the dotted rectangle are action-dependent). Making $p_0$ state-dependent would require more complicated machinery to verify that the game is an MPG (example through verifying condition C2) and would make the example unnecessarily complicated shifting away the attention from its main point. Nevertheless, in response to this and to another reviewer's comment, we provide an additional application of stochastic congestion games in the appendix which again involves state-dependent transitions.
> 3. We appreciate this idea and we would also like to expand on the techniques that we use in Section 4. However, given the multiple requests for presenting applications, we prefer to retain this section in the main paper. If the reviewer thinks otherwise, we can reconsider our decision.
>
> We hope that the above answer the reviewer's questions but we will be happy to reconsider parts of our paper if the reviewer thinks that this is necessary. We again thank the reviewer for their constructive feedback and their positive comments on our paper.

---

### Official Review · Reviewer_5gGX · 2021-10-31

**Correctness:** 4
**Technical Novelty And Significance:** 2
**Empirical Novelty And Significance:** 2
**Recommendation:** 5
**Confidence:** 3

**Main Review:**

Overall, this paper is well written. The theoretical analysis of this paper is presented clearly and seems to be rigorous.

My main concern is whether the so-defined MPGs are of enough interest. Specifically, in Prop $3.2$  (C1),  for an MDP to be an MPG, the state transition cannot be affected by the action, which is unrealistic in MARL setting. Second, in (C2), it remains unclear how to check whether the immediate rewards can be decomposed into the desired form with individual dummy terms. The authors only present a trivial example where $u_s^i$ is some constant $c^i$. Third, for cases of non-trivial state transition, the authors only provide a 2-agent MDP example in Figure 3. As a result, I think the practical applications of this paper are restricted.

**Summary Of The Paper:**

This paper studies Markov potential games (MPGs) that genralizes the classical normal form potential games. They show that each agent playing individual policy gradient guarantees the convergence to Nash policies with polynomially decaying error rate.

**Summary Of The Review:**

Although the theoretical analysis of this paper is well-organized and seems to be rigorous, the so-defined MPGs seem unrealistic in the MARL setting and are restrictive in practical applications.

---

> ### Author Response · Authors · 2021-11-19
> **Response to Reviewer 5gGX**
>
> We thank reviewer 5gGx for their positive comments and their feedback on our paper.
>
> Concerning the reviewer's concern, we appreciate the opportunity to clarify that Proposition 3.2 considers only the special class of MDPs that are potential at every-state. These are by far not all MDPs that are MPGs and in fact, as our Example 3 (Figure 3) shows, MPGs can even involve zero-sum interactions at some states. Moreover, as our calculations show (e.g., Example 2 in the Appendix), ordinal (and weighted) MPGs cover even broader and naturally motivated settings in which coordination is desirable. For these classes, our results still apply in an exact or asymptotic sense as we explain in Remark 1.
>
> At the very least, MPGs contain as an important sub-class, Markov games of pure common interests. As Dafoe et al (2020) outline, this is essentially the only class of cooperative AI interactions that has been fruitfully studied up to now (and which is still very actively studied as we indicate below). We also rephrased our Introduction to make this more clear.
>
> The interest in MPGs, in variations thereof or even in the clearly more restrictive setting of pure common interest Markov games is constantly growing. For reference, we provide a very indicative list of recent publications that study similar games
> - Learning in Nonzero-Sum Stochastic Games with Potentials, D. Mguni et al. ICML 2021, https://proceedings.mlr.press/v139/mguni21a.html
> - When Can We Learn General-Sum Markov Games with a Large Number of Players Sample-Efficiently? Song et al, 2021, https://arxiv.org/abs/2110.04184.
> - Decentralized Cooperative Multi-Agent Reinforcement Learning with Exploration, W Mao, T Başar, LF Yang, K Zhang, 2021, https://arxiv.org/abs/2110.05707.
> - Decentralized Cooperative Reinforcement Learning with Hierarchical Information Structure, H Kao, CY Wei, V Subramanian, 2021, https://arxiv.org/abs/2111.00781.
> - Independent Natural Policy Gradient Always Converges in Markov Potential Games, R Fox, S McAleer, W Overman, I Panageas, 2021, https://arxiv.org/abs/2110.10614.
> - Trust Region Policy Optimisation in Multi-Agent Reinforcement Learning.
> Jakub Grudzien Kuba et al, https://arxiv.org/abs/2109.11251.
>
> The large interest in MPGs is further showcased by the partially concurrent work by Zhang et al. (2021) who study precisely this class of games (and provide even more applications of MPGs, e.g., the medium access control game), the recent works by Valcarael-Macua et. al (2018) (ICLR), Mguni et al (2021) (ICML), and Mguni (2020), who study more restrictive variations of the current setup and the recently raised questions in Daskalakis et. al (2020) and Dafoe et. al (2020) about analyzing policy gradient in cooperative interactions beyond the case of identical interests (all these papers are referenced in the paper).
>
> We hope that these resolve the reviewer's concerns about the interest of the community in MPGs. Otherwise, we will be happy to further discuss this issue. We thank again the reviewer for giving us the opportunity to clarify this, and we hope that they will reconsider their score.

---

### Official Review · Reviewer_awwm · 2021-11-03

**Correctness:** 3
**Technical Novelty And Significance:** 3
**Empirical Novelty And Significance:** 2
**Recommendation:** 8
**Confidence:** 4

**Main Review:**

Strengths:
1. The definition of Markov potential games is known to be more delicate than that in the matrix (stateless) case. The authors have done a good job defining and giving (counter)-examples, in order to demonstrate this subtlety, mainly stemming from the introduction of the transition dynamics. This part, to my knowledge, has not been discussed in the literature before.
2. Both global convergence (for both the model-based and model-free setting) and sample complexity results (for the model-free setting) have been established, for finding the Nash equilibrium of the Markov potential game.


Weakness:
1. The global convergence, especially the exact case, is more-or-less expected, given Agarwal et al., 2020. In particular, given the opponent's policy being fixed, first-order stationarity => best-response, can be derived directly from that reference, which further leads to the definition of Nash equilibrium in the game setting. Hence, the novelty of this part, regarding optimization landscape and global convergence, is relatively limited.
2. It is claimed in the abstract that the convergence is "polynomially fast". I am not very sure about the accuracy and commonality of the wording. In fact, the $1/{\sqrt T}$ rate is not a fast one, but a standard one adapted from general nonconvex optimization.
3. Can the results in Theorem 4.4 be improved by choosing different parameters, e.g., alpha, eta, and T. It seems that $\epsilon^{-6}$ is not the standard rate in nonconvex stochastic optimization.
4. Some typos have been spotted, and some arguments are not accurate/professional:
1) Around Eq. (2), how the horizon length H is sampled (in fact, for unbiasdness, it should be sampled from a geometric distribution), is missed. Note that the setting in Daskalakis et al. (2020) is a finite-horizon one with random T (not an infinite horizon discounted one). So I am not sure if Lemma C.4's proof in this paper follows by just citing the results there directly. In fact, for infinite horizon discounted setting, the unbiasedness and bounded variance property of this estimator has been established before Daskalakis et al. (2020), see e.g., https://arxiv.org/pdf/1906.08383.pdf; https://arxiv.org/pdf/1807.11274.pdf.
2) The constants in the main theorems (and their dependences on the problem parameters) may need double-check.
5. How do the results compare with the very related recent work of https://arxiv.org/pdf/2106.00198.pdf? It might be worth discussing the differences between the two (thought I suppose it might be possible that the two works are concurrent).


**Summary Of The Paper:**

This paper carefully studies the definition of Markov potential games, and establishes the similarities and differences from the stateless case. It also shows the global convergence of independent policy gradient methods for Markov potential games, to the Nash policies.

**Summary Of The Review:**

In general, the paper is well-written, and I enjoy reading it. It has made valid contribution on multi-agent RL in potential games. With the minor comments above being addressed, I believe it can be a good paper deserving publication.

---

> ### Author Response · Authors · 2021-11-19
> **Response to Reviewer awwm**
>
> We thank reviewer awwm for the support of our paper and their constructive feedback.
> 1. The most important part in the proof of convergence is that Projected Gradient Ascent (PGA) on the potential function generates the same dynamics as each agent running independent PGA on their value function. The implication that first order stationarity implies best response is indeed straightforward and is part of Lemma D.3 which is relegated to the Appendix. We have rephrased our Abstract and Introduction to clearly reflect this. Thank you for pointing this out.
> 2. We mean that the convergence rate is polynomial in the approximation error. We rephrased our Abstract and Conclusions to avoid any confusion.
> 3. This is a great question for future work. We do not believe that the rate $O\left(\frac{1}{\epsilon^6}\right)$ is optimal. The $\alpha$-greedy parameterization is the reason behind the rate $O\left(\frac{1}{\epsilon^6}\right)$. Maybe adding a regularizer instead of using greedy parameterization (so that we avoid the boundary) or using variance reduction techniques can speed up the convergence rate. Nevertheless, we have chosen the parameters optimally for the $\alpha$-greedy parameterization; it turns out that $\alpha$ must of order $\epsilon^2$. We believe it is possible to get a rate of $O\left(\frac{1}{\epsilon^4}\right)$.
> 4. Thanks for pointing out the existence of some remaining typos. Upon an additional proofreading, we were able to find and correct some as the reviewer mentioned.
>    1. We had a related statement after Lemma 4.3. Since this was not clear enough or was coming too late in the text, we moved this statement after equation (2) and expanded it with an explicit mention to the geometric distribution. We thank the reviewer for pointing out that unbiasedness was already established in the two mentioned papers which we were not aware of. We now cite those papers appropriately. Please also note that infinite horizon with discounted factor $\gamma$ is effectively a finite horizon in which a MDP has $1-\gamma$ probability to terminate and $\gamma$ to continue (i.e., geometric distribution). This is how we use it to get a sample for the gradient of each agent's value function.
>    2. We double-checked the constants in Theorems 4.2 and 4.4 and found no issues. However, in response to another reviewer's comment, we rephrased both Theorems to reflect that the step-size does not require knowledge of the horizon.
> 5. The two papers indeed study MPGs in parallel which showcases the growing interest of the ML and AI community to this important class of games. The sample-complexity analysis in that paper is subsequent to our results but the sample-based algorithm is developed from a different perspective. In particular, their learning method exploits an averaged MDP which can be viewed as a model-based policy evaluation method (in contrast to our model-free estimation). Interestingly, both approaches lead to the same sample complexities and as the authors of that paper point out, it will be useful to study if this dependence is fundamental or not. In our revised version, we acknowledge the above in the Other Related Works subsection.
>
> We hope that the above actions address all issues raised by the reviewer but we are happy to provide additional inputs if this is necessary. We thank again reviewer awwm for their detailed feedback and positive comments on our paper.

---

### Official Review · Reviewer_43Uo · 2021-11-05

**Correctness:** 4
**Technical Novelty And Significance:** 3
**Empirical Novelty And Significance:** 2
**Recommendation:** 6
**Confidence:** 3

**Main Review:**

Strength:

1. The paper did a great job introducing the MPG and showing us intuitions about the game with examples and observations.
2. The paper proved the convergence to $\epsilon$-NE and provided the convergence rate for learning agents following independent policy gradient or stochastic gradient in MPG.
3. The paper additionally provides the empirical results to verify the convergence.
4. The paper describes several meaningful open questions and future directions that I find quite valuable.

Weakness:

1. I wish to see more MPG instances (e.g., with potential function of different smoothness or convexity) in the empirical evaluation and wonder how the performance of multiagent learning differs in different games. I think it is also possible to include classical RL algorithms and see how they perform compared to policy gradients.
2. The paper could have a much more profound technical contribution if it could touch upon any of the open questions mentioned in the end.

Questions:

1. Is it possible to generalize the theorem such that the algorithm does not need to know the number of iteration beforehand, in analogy to the doubling trick in bandit literature?
2. Is it possible to generalize the game setting to infinity state setting using the techniques from literature [1] of linear Markov decision process? It seems to me the MPG definition can be further generalized in this way.

[1] Jin, Chi, et al. "Provably efficient reinforcement learning with linear function approximation." Conference on Learning Theory. PMLR, 2020.


**Summary Of The Paper:**

The paper introduces the Markov Potential Game (MPG), which generalizes the classical potential game. The authors then point out several properties of MPG. It proves the convergence to $\epsilon$-NE for learning agents following independent policy gradient or stochastic gradient. The paper also presents the experiment on a MPG of congestion games that verifies the theoretical results.

**Summary Of The Review:**

The paper is overall well-written and established the preliminary studies of MPG problem and multi-agent learning algorithm for convergences, though there are still many unsolved questions in MPG such as Price of Anarchy. Therefore, I recommend a weak-accept of this paper.

---

> ### Author Response · Authors · 2021-11-19
> **Response to Reviewer 43Uo**
>
> We thank reviewer 43Uo for their time to read our paper and for their positive comments.
> 1. Concerning the reviewer's first question, we clarify that our algorithms do not need to know the number of iterations beforehand. To see this, observe that the step-sizes in both main Theorems do not make use of the time horizon T, but rather only depend on the parameters of the game. To phrase this differently, for the given step-size, $\eta$, Theorems 4.2 and 4.4 state that after at most $T$-many iterations (where $T$ depends on $1/\eta$) convergence will be reached. The doubling-trick can be very useful in extensions of the current setting (where one may need a diminishing step-size to achieve convergence), and we thank the reviewer for this suggestion. To avoid creating this ambiguity also to other readers, we rephrased both Theorems 4.2 and 4.4. We hope that the current formulation reflects the above more clearly and we thank the reviewer for bringing this up.
> 2. Concerning the reviewer's second question about the use of techniques from linear Markov processes to expand the current results, we comment that this is indeed a very smart idea for future work. Our initial efforts suggest that this is a non-trivial direction. We added it in our open questions section along with a reference to [1]. We thank the reviewer for this insightful comment.
> 3. In the revised version, we also address head-on the first weakness mentioned by the reviewer. In the appendix, we provide another setting of MPGs, called stochastic potential games, and conduct an experiment with independent policy gradient. Such games have considerably more states than the ones in Section 5 and independent policy gradient requires more iterations before reaching convergence.
> 4. Concerning the performance of multi-agent learning in different games, we remark that the variations of the main experiment in the appendix aim to address precisely this concern. Our results show that convergence is a robust phenomenon since it also holds (provably) for weighted MPGs, asymptotically for ordinal MPGs and (experimentally) also for variations thereof. Going beyond MPGs (ordinal or weighted MPGs) to mixed-motives Markov games is the current frontier.
> 5. Concerning the weakness that the reviewer mentions, we agree that the open questions of the conclusions and the convergence of other RL algorithms are indeed very interesting and impactful directions for future work. Our follow-up research in one of these directions (which we cannot cite here to preserve anonymity) shows that reasonably addressing any of these questions (e.g., convergence of variations of independent policy gradient) most likely requires a standalone paper.
>
> We hope that our responses address the reviewer's questions but we will be happy to offer any additional input if required. We thank the reviewer for the valuable interaction and their positive comments for our paper.

---

### Author Response · Authors · 2021-11-19
**We thank the reviewers - our updates are in blue**

We thank all reviewers for their time to read through our paper and for their valuable feedback. All changes that we did in response to the reviewers' comments are highlighted in blue in the revised manuscript. We hope that these updates address the reviewers' suggestions and we thank them again for their positive comments and valuable inputs.

---

### Decision · Program_Chairs · 2022-01-20

**Decision:**

Accept (Poster)

**Comment:**

This paper considers an important problem, Multi-Agent Reinforcement Learning, and looks at a subclass of problem, the Markov Potential Game.

Even though this class is not the more generic one (as pointed out by a reviewer), one must start somewhere before (and maybe the results cannot easily or at all be extended to a larger class), so I will not personally take this as a strong negative concern.

The other reviewers are rather positive about the result and the techniques, and I concur with them.

I will therefore recommend acceptance.